



# Analysis of 3D Cloud Effects in OCO-2 XCO2 Retrievals

Steven. T. Massie[1], Heather Cronk[2], Aronne Merrelli[3], K. Sebastian Schmidt[1], Hong Chen[1], and David Baker[4]

[1]Laboratory for Atmospheric and Space Physics, University of Colorado, Boulder, Colorado, 80303, USA
[2]Colorado State University, Fort Collins, Colorado, 80523, USA
[3]Space Science and Engineering Center, University of Wisconsin-Madison, Madison,
Wisconsin, 53706, USA
[4]Cooperative Institute for Research in the Atmosphere, Colorado State University, Fort Collins, Colorado, 80523, USA

*Correspondence to:* Steven T Massie (Steven.Massie@lasp.colorado.edu)

**Abstract.** The presence of 3D cloud radiative effects in OCO-2 retrievals is demonstrated from an analysis of 2014-2019 OCO-2 XCO2raw retrievals, bias corrected XCO2bc data, ground based Total Carbon Column Observation Network (TCCON) XCO2, and Moderate Resolution Imaging Spectroradiometer (MODIS) cloud and radiance fields. Averaged over
the year, 40 % and 75 % of OCO-2 Quality Flag QF=0 (best quality) and QF=1 (lesser quality) retrievals are within 4 km of clouds. 3D radiative transfer calculations indicate that 3D cloud radiative perturbations at this cloud distance, for an isolated low altitude cloud, are larger in absolute value than those due to a 1 ppm increase in $CO_2$. OCO-2 measurements are therefore susceptible to 3D cloud effects. Four 3D cloud metrics, based
upon MODIS radiance and cloud fields and stand-alone OCO-2 measurements, relate XCO2bc-TCCON averages to 3D cloud effects. This analysis indicates that the operational bias correction has a non-zero residual 3D cloud bias for both QF=0 and QF=1 data. XCO2bc –TCCON averages at small cloud distances differ from those at large cloud distances by -0.4 and -2.2 ppm for the QF=0 and QF=1 data over the ocean. Mitigation of
3D cloud biases by a Table look-up technique, that utilizes nearest cloud distance (Distkm) and spatial radiance heterogeneity (CSNoiseRatio) 3D metrics, reduces QF=1 ocean and land XCO2bc –TCCON averages from -1 ppm to near ± 0.2 ppm. The ocean QF=1 XCO2bc-TCCON averages can be reduced to the 0.5 ppm level if 60 % (70 %) of the QF=1 data points are utilized, by applying Distkm (CSNoiseRatio) metrics in a data screening
process. Over land the QF=1 XCO2bc–TCCON averages are reduced to the 0.5 (0.8) ppm level if 65 (63) % of the data points are utilized by applying Diastkm (CSNoiseRatio) data screening. The addition of more terms to the linear regression equations used in the current bias correction processing, without data screening, however, did not introduce an appreciable improvement in the standard deviations of the XCO2bc-TCCON statistics.

## 1 Introduction

The Orbiting Carbon Observatory (OCO-2) measures the column-averaged atmospheric $CO_2$ dry air mole fraction, referred to as XCO2, on a global basis (Eldering et al., 2017).
Space based measurements of XCO2 can improve our understanding of surface $CO_2$ fluxes if XCO2 variations are accurately measured to the 0.3 % level (~1 ppm) on spatial scales





from less than 100 km over land and ~1000 km over the ocean (Rayner and O'Brien, 2001; OCO-2 L2 ATBD, 2019).

OCO-2 derives XCO2 from an optimal estimation methodology (Rodgers, 2000) that is applied (O'Dell et al., 2018) to spectra in three spectral bands: the 0.76 µm $O_2$ A-band, the 1.61 µm weak $CO_2$ band, and the 2.06 µm strong $CO_2$ band. The spectral resolutions of the three spectrometers are greater than 19,000 and are sufficient to resolve molecular pressure-broadened lines. Each spectral band is comprised of 1016 wavelength samples. The retrieval includes a state (solution) that includes $CO_2$ at 20 levels, surface pressure,

$H_2O$ and temperature profile scale factors, aerosol and cloud opacity, land or ocean surface albedo, and spectral dispersion shifts. To boost signal to noise over the dark ocean surface, XCO2 measurements over the ocean rely on sun-ocean-sensor glint viewing geometry. Measurements over land are collected in nadir or glint view geometry.  A third mode, target mode, commands OCO-2 to observe many points around a specific targeted area. In this

mode the sensor azimuth and zenith angles vary appreciably for a given surface location, which is not the case for the glint and nadir modes.

Clouds and aerosols definitely complicate the radiative transfer associated with the OCO-2 measurements. Connor et al. (2016) identify aerosols (solid and liquid particles) as the most important error source, followed by spectroscopic and instrument calibration

uncertainties. To minimize the influence of clouds, the cloud pre-processor (Taylor et al., 2016) applies two fast algorithms to screen for clouds. The "A-band Preprocessor" solves for the surface pressure assuming that no clouds or aerosols are present. Differences greater than 25 hPa between retrieved and a priori surface pressure lead to the exclusion of a profile from the Level-2 "Full Physics" operational retrieval (OCO-2 L2 ATBD, 2019). The

second algorithm compares column-integrated $CO_2$ from the weak and strong $CO_2$ bands. If the ratio of the $CO_2$ columns deviates significantly from unity, then the profile is excluded from the Full Physics retrieval. The preprocessors are very efficient, but they do not catch all cloudy scenes, especially if there are low altitude clouds present. Of the 1 million measurements made each day, ~25 % pass the preprocessor filters and enter the

operational retrieval (O'Dell et al., 2018).

Primary validation of OCO-2 XCO2 relies upon comparison to the Total Carbon Column Network (TCCON) ground based measurements of XCO2 (Wunch et al., 2017). Twenty-seven TCCON stations (see http://tccon.caltech.edu) utilize Fourier Transform Spectrometer instrumentation. TCCON observation geometry is direct solar viewing, and

the XCO2 measurements are accurate to 0.5 ppm (Wunch et al, 2010). Comparisons of XCO2raw (the XCO2 that is produced by the operational retrieval) to TCCON measurements reveal that TCCON measurements are approximately 1 ppm larger than XCO2raw values, as discussed in the Version 9 Data Product User's Guide (2018). Based upon these and other comparisons, the OCO-2 algorithm team applies multi-variable linear

regressions separately over land and ocean to bias correct the XCO2raw retrievals to XCO2bc values. The variables in the bias correction equations include differences in the retrieved and a priori surface pressures, the sum of aerosol optical depths for large aerosol particles (for land data), and a "CO2graddel" term. CO2graddel is a measure of the difference in the vertical gradients of the a priori $CO_2$ and retrieved vertical profiles (see

Eq. (5) of O'Dell et al., 2018).

Not all physics, however, is included in the Full Physics retrieval. The subject of this paper is 3D cloud effects. The operational retrieval is a 1D-column retrieval, by necessity.





The computer processing of a single profile takes several minutes. More than 100,000 profiles are retrieved per day, requiring an appreciable amount of computer processing. With regard to 3D cloud effects, radiances from a clear sky footprint may be perturbed by a cloud several kilometers from the clear sky footprint. The 1D retrieval, however, uses the independent pixel approximation, by which radiative transfer optical properties are those within a single 1D column. The 1D retrieval does not consider the radiative effects of clouds outside of the 1D column. The operational retrieval iterates for the state vector elements of the surface pressure, aerosol, surface reflectance, and the $CO_2$ vertical profile that minimizes the differences in the observed and forward model spectra. The state vector elements frequently take on unrealistic values in the converged solution.

Previous papers have demonstrated the presence and effects of 3D cloud effects in other experiments and the OCO-2 experiment. Várnai and Marshak (2009) demonstrated that MODIS reflectance at various wavelengths between 0.47 and 2.12 µm increases as cloud distances decrease at cloud distances less than 10 km, and that the effect is strongest at shorter wavelengths. Okata et al. (2017) modeled 3D cloud effects, finding positive 3D–1D radiance differences, for solar zenith angles greater than 5°, for periodic cuboid clouds of 2.5 km height. Merrelli et al. (2015) applied the SHDOM 3D radiative transfer code, and the OCO-2 retrieval code, and concluded that the OCO-2 cloud-screening algorithm had difficulty in rejecting clouds that filled less than half of the field of view. Retrieved XCO2 were offset low from clear sky retrievals by 0.3, 3, and 5-6 ppm for soil, vegetation, and snow surfaces. Massie et al. (2017) analyzed version 7 OCO-2 XCO2 in conjunction with MODIS radiance fields, demonstrating that XCO2 decreased as a cloud-radiance field inhomogeneity metric increased in target mode observations. Here we extend Massie et al. (2017) by analyzing additional 3D cloud metrics, and relate each of the metrics to the global set of TCCON XCO2 measurements obtained from 2014 through 2019.

Our study is organized in the following manner. In Section 2 we discuss the OCO-2, Moderate Imaging Spectroradiometer (MODIS), and TCCON data that is analyzed. Details of the bias correction procedure are presented in Section 3. We define four 3D metrics that are derived from MODIS-based files (such as nearest cloud distance) and *stand-alone* OCO-2 metrics in Section 4. We compare the utility and effectiveness of the MODIS and stand-alone metrics, since the stand-alone metrics are readily calculable from the OCO-2 data files, while the MODIS-based files impose an additional level of processing complexity. In Section 5 we demonstrate that over half of the OCO-2 measurements are within 4 km of clouds, and demonstrate in Section 6 that the 3D cloud effect over ocean and land has a larger radiative perturbation (in absolute terms) at this cloud distance than perturbations for a 1 ppm increase in XCO2. Distributions of XCO2raw –TCCON and XCO2bc – TCCON are related to the four 3D cloud metrics in Section 7. We demonstrate that 3D cloud biases in XCO2bc – TCCON remain after the current bias correction processing for both Quality Flag QF=0 (best quality) and QF=1 (lesser quality) data. While Section 7 focuses on global analyses, we demonstrate in Section 8 that the 3D effects appear readily in local scenes. Mitigation of the 3D cloud biases by application of a Table look-up correction is discussed in Section 9. Mitigation of the 3D cloud biases by data screening by the four 3D metrics is investigated in Section 10. Mitigation by adding terms to the current bias correction equations, without data screening being applied, is discussed in Section 11. Finally, Section 12 summarizes the findings of the previous sections.





## 2 Data


OCO-2 product files are available from the NASA Earthdata website (https://earthdata.nasa.gov/). Level 2 L2Std (standard) and L2Dia (diagnostic) files contain retrieved XCO2 (referred to as XCO2raw data). "Lite" files contain the XCO2raw and biased corrected XCO2bc data, with one file containing all converged retrievals for one day. The Quality Flag (QF) is set to 0 for the best quality data, and to 1 for lesser quality

data. Each OCO-2 measurement has an associated 16 digit Sounding ID that uniquely identifies each XCO2 profile. Over 100,000 successful retrievals are contained in a single daily Lite file. We focus upon Version 9 and 10 OCO-2 data files in our study, with the majority of presented figures and tables based upon the Version 10 data. The Version 10

data we analyze is derived from "beta" release files, housed at JPL, prior to the formal release to the Earthdata GES DISC archive.

Auxiliary files (Cronk et al., 2018), not archived by the NASA Earthdata file system, contain MODIS radiances at 500m spatial resolution, cloud mask, cloud fraction, cloud optical depth, and geolocation (based upon OCO-2 Version 9 data), matched to the OCO-

2 Sounding ID. We refer to these files as Colorado State University "CSU files". MODIS and OCO-2 fly in formation in the NASA "A-train", with OCO-2 flying six minutes in front of MODIS Aqua. For each Sounding ID there are MODIS data points within 50 km east and west of the OCO-2 observation point. In relation to each OCO-2 observation footprint, we determine the closest MODIS field point for which the MODIS cloud mask

indicates a cloud, or for which the MODIS cloud optical depth is greater than unity. Knowing the geolocation positions of these two points, the distance in km between the footprint and cloud, and the angle between the observation footprint and cloud, are calculated. 3D cloud effects likely are dependent upon the distance of a cloud to the observation footprint and sun-cloud-footprint viewing geometry considerations.

In addition to the OCO-2 and MODIS-based data, our analyses includes data files that combines this data with adjacent TCCON measurements. We refer to these files as "Validation" files. A TCCON measurement is associated with an OCO-2 measurement, on the same day, if the difference in geolocation is less than 2.5° in latitude and 5° in longitude. These files allow us to calculate the statistics associated with XCO2bc-TCCON and

XCO2raw-TCCON comparisons over ocean and land. Table 1 lists the TCCON sites and data used in our analyses. Wunch et al. (2015) discusses the TCOON data version we analyze.

We also examine differences in averaged OCO-2 spectra, as a function of distance from nearest clouds and as a function of XCO2bc to illustrate the perturbations in radiance that

are due to 3D cloud effects. OCO-2 spectra are contained in the level 2 diagnostic (glint oco2_L2DiaGL.. and nadir oco2_L2DiaND..) files. For the spectral analysis we co-process the diagnostic, Lite, and CSU MODIS files.

For the determination of the standard deviation of the radiances for adjacent observation footprints, which is used to determine the H(Continuum) 3D metric discussed

in Section 4, we analyze the $O_2$ A-band continuum radiances that are archived in the OCO-2 Version 10 Lev1b files (glint oco2_L1bScGL.. and nadir oco2_L1bScND..) files. The Lev1b Version 9 files also contain "colorslice" data which is used to define the CSNoiseRatio discussed in Section 4.


## 3 Bias correction procedure

As discussed by O'Dell et al. (2018) and in the Version 9 OCO-2 Data Product User's
Guide (2018, see Table 3.4), the bias correction procedure compares Level 1 retrieved
XCO2raw and TCCON XCO2, and produces *bias corrected* XCO2bc values, based upon
the following equations for ocean glint and land nadir Version 9 observations.

$$XCO2bc = (XCO2raw – Foot(fp) – Feats) / TCCONadj. \qquad (1)$$

For ocean glint observations,

$$Feats = - (0.245 * dPsco2) + (0.09* (CO2graddel + 6.0)). \qquad (2)$$

For land nadir observations,

$$Feats = - (0.90 * dPfrac) – (9.0*DWS) – (0.029 * (CO2graddel -15.0)). \qquad (3)$$

The *footprint bias* Foot(fp) for footprints (fp) 1 through 8 varies monotonically from -0.36
to 0.34. The Version 9 TCCONadj values are 0.9954 and 0.9953 for land and ocean
observations. dPsco2 is the difference (in hPa) between the retrieved and a priori surface
pressure evaluated at the strong $CO_2$ band geographic location, while dPfrac (in ppm units)
is

$$dPfrac = XCO2raw * (1.00 – Papriori/ Pretrieved). \qquad (4)$$

For Version 9 and 10 data the Papriori is taken from the GEOS-5 Forward Processing for
Instrument Teams (GEOS-FP-IT) analysis.  CO2graddel is a measure of the difference in
the retrieved and prior CO2 vertical gradient, and is applied in Eq. (2) if CO2graddel is less
than -6.0. DWS is the sum of the vertical optical depths of the dust, water, and seasalt
aerosol components.
     For Version 10 data Eq. (2) still applies, but with dPsco2 and CO2graddel coefficients
of 0.213 and 0.0870, and TCCONadj equal to 0.995 (Version 10 OCO-2 Data Product
User's Guide (2020), see Table 3.3). For land observations,

$$Feats =  – (0.855 * dPfrac)  –  0.335 * (max(logDWS,-5) + 5.0)$$
$$– (0.0335 * (CO2graddel -5.0))  +  5.20 (AODfine -0.03), \qquad (5)$$

where AODfine is the fine aerosol optical depth (sulfate plus organic carbon aerosol), and
TCCONadj is equal to 0.9959. The Version 10 and 9 Foot(fp) values differ slightly.
In the application of Eqns. (1) – (3), the retrieval provides dPsco2, dPfrac, DWS, and
CO2graddel bias correction values that are used in the bias correction calculations. The
XCO2raw values are designated as QF=0 or QF=1 data points from a series of exceedance
checks on many variables, including the bias correction variables. The operational bias
correction only uses the QF=0 data points to determine the linear coefficients in Eqns. (2)
and (3).



The differences in XCO2raw and XCO2bc are due to several factors. First of all, there are uncertainties in the spectroscopic parameters (line strengths, pressure broadening coefficients, energy levels, and specifications of the molecular line shape, including line-mixing complications). Calibration errors, especially in regard to the instrument line shape, are also important. Incorrectly modeled physical scene characteristics, such as errors in the aerosol single scattering property or surface BRDF specification, and/or 3D cloud scattering considerations, also have influence upon the XCO2raw and XCO2bc differences.


The operational retrieval, however, does not include 3D cloud effects. We will calculate 3D cloud metrics based upon the MODIS files and "stand alone" OCO-2 data, and investigate if application of the 3D metrics in a Table look-up correction, or by data screening by the 3D metrics, leads to a reduction in the standard deviations and averages of TCCON-XCO2bc probability distribution functions (PDFs). We also add 3D cloud metric terms to the bias correction Eqns. (1)-(3) to determine if they reduce TCCON-XCO2bc standard deviations and averages.



**4 Metrics**

Several metrics are analyzed in this paper: a) nearest cloud distance (abbreviated as Distkm), and the sun-cloud-footprint scattering angle, b) MODIS radiance field H(3D), c) CSNoiseRatio, and d) OCO-2 footprint radiance standard deviations, H(Continuum). Metrics a) and b) are calculated from analyses of the CSU files, while metrics c) and d) are based upon *stand-alone* OCO-2 data. We will apply all of the metrics in subsequent sections of this paper, and compare how well each metric performs in reducing the scatter in the TCCON-XCO2bc standard deviations and averages over ocean and land.



The CSU files are processed to determine the distance in km of the OCO-2 Lite file observation data points to the nearest MODIS cloud. The distance is simply the hypotenuse of the triangle formed by the difference in latitude and longitude of the center of the OCO-2 footprint and the nearest MODIS cloud, with the longitude difference multiplied by the cosine of the latitude. The sun-cloud-footprint scattering angle is the angle between the sun to nearest cloud vector and the nearest cloud to observation footprint vector. The Distkm metric frequently refers to clouds that are *outside* of the geospatial scan pattern defined by the eight OCO-2 observation footprints. The Distkm metric cannot be specified from OCO-2 observations.



The H(3D) metric (Liang, Di Girolamo, and Platnick, 2009; Massie et al., 2017), as applied to the radiance field,

H(3D, kcir) = standard deviation of the Radiance field / average of Radiance field,    (6)


is a measure of the *inhomogeneity* of the radiance field, and is calculated from the CSU file radiance fields. For a cloudless scene with no surface reflectance variations, the H(3D) parameter approaches zero, while for scenes with broken cloud fields or surface reflectance heterogeneity, the H(3D) metric is larger. The H(3D, kcir) values are calculated for four averaging circle radii (kcir) of 5, 10, 15 and 20 km, that surround each OCO-2 footprint. 95 % of the H(3D) values vary between 0.0 and 0.80 over the ocean and between 0.0 and 0.66 over land. The 10 km circle H(3D) data is used in our study. Figure 1 of Várnai and






Marshak (2009) indicates that MODIS reflectance at wavelengths between 0.47 and 2.12 µm increased (i.e. that 3D cloud effects are present) for cloud distances less than 10 km, with nearly zero increase in reflectance at larger distances. We find that there is a larger
inhomogeneity in the radiance field over the ocean than over the land. The H(3D) metric increases as cloud inhomogeneity increases.

The OCO-2 CSNoiseRatio uses the sub-footprint spatial information contained within the "colorslice" data. As discussed by Crisp et al. (2017, see their Fig. 2), each of the 8 footprint samples are an average of 20 pixels. For a subset of 20 columns (the spectral
dimension), the individual pixel level data is returned from the instrument and stored as "colorslices" in the level 1 L1b data files. The specific 20 columns are chosen at specific spectral locations in each of the OCO-2 bands, primarily to support the de-clocking algorithm. Each band contains 5 or 6 colorslices at continuum wavelengths. The spatial mean and standard deviation are computed for each of these continuum colorslices, and
then the final mean and standard deviation for that individual sounding is computed across those 5 to 6 values. Computing a median over the available continuum slices makes the calculation robust to isolated bad pixel values, which can be caused by cosmic ray hits on the detectors. The "CSNoiseRatio" used in this paper is the ratio of the continuum radiance spatial standard deviation and the noise level at the continuum radiance level as predicted
from the radiometric noise model. The CSNoiseRatio has an expected value of unity if the continuum radiance in the footprint is spatially constant, as the standard deviation across the pixels should be due to the detector noise. The CSNoiseRatio values increase as the within-footprint radiance inhomogeneity increases. Note that each observation footprint has an extent of approximately 1.3 km (cross-track) by 2.3 km (along-track) at the Earth's
surface. The CSNoiseRatio values increase as cloud inhomogeneity, within and/or outside of each observation footprint, increases.

Finally, the H(Continuum) metric is calculated from Eq. (7), based upon the observed radiance Radobs at a specific footprint, and the standard deviation of the radiance field, with radiances given by the OCO-2 $O_2$ A-band level1b *continuum radiances*.


H(Continuum) = 100 (standard deviation of  the Radiance field / Radobs ),          (7)

For a specific observation footprint, we focus upon the primary west to east row of eight footprints that contains the specific footprint, and two adjacent rows, one north and one
south of the primary row (see Fig. 9, discussed below). There are therefore 23 adjacent footprints that we associate with a specific footprint. For each specific footprint, the 23 adjacent footprint continuum radiances are included in each H(Continuum) calculation. All footprints are given equal weight in applying Eq. (7), including footprints 1 and 8 (the edge footprints). 95 % of the $O_2$ A-band H(Continuum) values vary between 0 and 24 over the
ocean, and between 0 and 27 over land. H(Continuum) increases as cloud inhomogeneity increases.

**5 The proximity of OCO-2 observations to clouds**

Figure 1 presents the fraction of Lite file glint and nadir observations that have a cloud within a circle of a specified radius in km, in summer for five 20º latitude bands, for 2014 - 2019. The calculations utilize distance bins from 0 to 35 km, with fractions normalized



to 100 % for the 35 km circle radius. In average terms, 40 % and 73 % of the observations over the ocean and land are within 4 km of clouds for the QF=0 and QF=1 cases, respectively. The tropical 0º-20º and -20º-0º latitude bands have observations that are closest to clouds. This is of importance since the tropics have relatively few OCO-2 observations, compared to other latitudinal bands. Carbon cycle fluxes in the tropics are large and are very important in regards to understanding the global carbon cycle.

Table 2 presents the fraction of observations that have a cloud within 4 km of an observation for each season. The minimum and maximum values for the four seasons are in the 21-58 % and 55-96 % ranges for the QF=0 and QF=1 cases. Averaged over the year, 40 % and 75 % of the QF=0 and QF=1 observations are within 4 km of a cloud. Fig. 1 and Table 2 indicate that OCO-2 QF=1 data is appreciably closer to clouds than the QF=0 data. The QF=1 data is therefore more susceptible to 3D cloud effects than the QF=0 data.

## 6 Radiative Transfer Sensitivity Calculations

To illustrate the relative sensitivity of glint and nadir observations to 3D cloud effects, we applied the Spherical Harmonic Discrete Ordinate radiative transfer Method (SHDOM) 3D radiative transfer code to the same sparse cloud scene, varying glint and nadir viewing geometry and other parameters (surface reflectance). This cloud scene is illustrated below in Fig. 9. SHDOM (Evans, 1998; Pincus and Evans, 2009) is applied by specifying a 3D model atmosphere with a specified 3D field of cloud optical properties. Radiation fields at satellite altitude for 1D column (independent pixel approximation, IPA) and 3D mode are calculated separately. Comparison of the IPA and 3D calculations then indicates the size of the 3D cloud effect radiative perturbations.

Figure 2 presents SHDOM radiative perturbations for all three OCO-2 bands, based upon the atmospheric base-state and perturbed parameters given in Table 3, with monochromatic total optical depth on the x axis and radiative perturbations on the y axis. Perturbations are applied individually one at a time, e.g. for the calculation of the partial derivative of radiance with respect to a change in surface pressure, all other variables are kept at their base state values. The base state $CO_2$ is 400 ppm at a surface pressure of 1016 hPa.

The cloud field is derived from the MODIS 250 m radiance field on June 12, 2016 over the ocean (and graphed in Fig. 9). As discussed by Massie et al. (2017), the MODIS cloud mask does not identify all clouds that are visible in MODIS imagery (available from the NASA Worldview website https://worldview.earthdata.nasa.gov/). MODIS 250 m field radiance and MODIS cloud mask data can be used together to generate a cloud field that includes cloud elements not identified by the MODIS cloud mask. The SHDOM cloud field assigns a cloud to a location if the MODIS radiance at that location is greater than or equal to scene-specific MODIS radiance thresholds. The scene-specific radiance thresholds are calculated from the radiances at scene locations in which the cloud mask indicates a cloud, and/or when the MODIS cloud optical depth is greater than unity. The cloud height is set at 1.8 km. This is the median height of the PDF of trade wind cumuli heights determined from an analyses of 30m Advanced Spaceborne Thermal Emission and Reflection (ASTER) stereo data (Genkova et al. 2017). This is also the cloud height used by Massie et al. (2017) in their 3D calculations for an OCO-2 target mode observation centered over the Lamont, Kansas TCCON site.





The 1D calculations are perturbed (see Table 3) individually by 10 hPa and 10 ppm for
surface pressure and $CO_2$ perturbations, and by surface reflectance (for nadir) or surface
wind (for glint), and aerosol optical depth perturbations. Aerosol optical depth vertical
structure is the same for all x-y grid points, but the total aerosol optical depths are equal to
e.g. 0.11 and 0.165 for the base and perturbed state $O_2$ A-band calculations The OCO-2
ABSCO database of molecular line cross sections (Payne, 2016) is used to specify the gas
optical depth structure in the x, y, z 3D grid (of size 32 km x 32 km x 30 km, with a
horizontal grid cell size of 0.5 km x 0.5 km).  SHDOM was applied in monochromatic
calculations at 17 wavelengths, in which the total gas plus aerosol optical depth ranges
from small to large values, for Lambertian surface scattering over land and Cox-Munk
surface wind dependent bidirectional diffuse reflectance (BRDF) over the ocean.
The curves labeled as "3D" in Fig. 2 are percent differences between the 3D and 1D
calculations, for base state conditions, at 4 km west of a typical cloud in the MODIS cloud
field (with the sun along the negative x axis at a solar zenith angle of 20°). The other curves
are 1D perturbations, normalized to the stated perturbation amount. For example, the "1
ppm CO2" curve is derived by dividing the SHDOM radiance field differences for the 400
and 410 ppm conditions by 10. The 1D curves are radiance perturbations also at 4 km from
the cloud, and since the 1D column calculation does not have any knowledge of nearby
clouds, the 1D curves are not influenced by nearby clouds. All of the panels in Fig. 2 have
x-axes expressed in terms of the gas plus aerosol vertical optical depths of the base state
atmosphere. 3D radiative perturbations are largest at small optical depths, while 1 ppm $CO_2$
perturbations are largest at large optical depths. This indicates that 3D cloud effects impose
spectral perturbations with an optical depth structure that differs from $CO_2$ mixing ratio
perturbations.
Figure 2 indicates that a cloud 4 km away from a clear sky footprint has 3D cloud effect
radiative perturbations in the WCO2 and SCO2 bands that are larger at small optical depths
than a 1 ppm $CO_2$ perturbation. The WCO2 (SCO2) perturbations are near 2.1 % (1.5 %)
and 1.4 % (1.0 %) for the glint and nadir cases, while the "1 ppm CO2" curves have values
less than 1 % in absolute value. This comparison is relevant since the observational goal
of OCO-2 is to measure XCO2 to 1 ppm accuracy on regional scales. OCO-2 observations
therefore are susceptible to 3D cloud effects.
From a radiative transfer perspective, Fig. 2 indicates that ocean glint observations are
more susceptible to 3D cloud effects than land nadir observations. Since Fig. 1 and Table
2 indicates that clouds are closer to observations over the ocean than over land, the Fig. 1
and 2 calculations, in combination, indicate that 3D cloud effects are likely more prevalent
for the ocean glint measurements.
The Fig. 2 calculations are not influenced by cloud shadows, since the observation point
is west of the cloud position. While Fig. 2 focuses upon radiative perturbations away from
a cloud, 3D cloud effects also include cloud shadows, which decrease the sensed radiances.
It is expected that radiance enhancements and radiance dimming both occur in OCO-2
observations, which can yield both negative and positive XCO2 variations to the local
scene.
It is expected that viewing and scattering geometry play an important role in 3D cloud
effects. Liquid and ice particles have phase functions which have dominant forward
scattering peaks, and the scattering of solar photons off of the side of a cloud is an important
component of the 3D cloud effect. Figure 3 illustrates the angular dependence of 3D cloud





effects along a circle of 4 km radius that surrounds an isolated cloud. The calculations refer
to a continuum wavelength with the smallest possible gas optical depth. Observation
footprints are to the west, north, east, and south of the cloud at angles of 0º, 90º, 180º, and
270º, with the sun at the 0° angle along the negative x-axis and the sensor along the positive
x-axis. There is a factor of two variation, as a function of the location of the observation

footprint, in the 100 (3D-IPA) / IPA values. The largest values occur when the observation
footprint is west of the cloud (angle=0°). The solar beam scatters off of the west side of the
cloud back to the observation footprint, which is followed by additional scattering off of
the surface towards the sensor along the positive x axis. This solar beam side-of-cloud
scattering contribution does not take place when the observation footprint is east of the

cloud (angle=180°), so the 3D effect is then smaller.
        Since the OCO-2 cloud screening preprocessor frequently does not reject scenes with
a few low altitude "popcorn" clouds, the metrics of nearest cloud distance and the sun-
cloud-observation footprint scattering angle are useful rudimentary metrics to characterize
a cloud scene. But they do not characterize completely a cloudy scene with numerous

clouds. As more and more clouds are added to a scene that surrounds an observation point,
there is a complicated interaction of perturbative effects from the individual clouds

**7 Global statistics**

The Validation files reveal the dependencies of XCO2bc-TCCON and XCO2raw-TCCON
upon the various 3D metrics. Fig. 4 presents contour maps of the number of XCO2raw-
TCCON and XCO2bc-TCCON observations over the ocean versus nearest cloud distance.
There are more data points at smaller than at larger cloud distances, especially for the QF=1
data. The bias correction moves the center of the XCO2raw-TCCON distributions upwards

towards the XCO2bc-TCCON = 0 line, especially for the QF=0 data. This is not as apparent
for the QF=1 distributions, keeping in mind that QF=1 data is not used in the operational
bias correction calculations.  For the 0 to 2 km cloud range there is a noticeable asymmetry
in the QF=1 distributions, with a "tail" of negative XCO2bc-TCCON data points. This is
visually apparent by following the aqua-marine-blue contour line from larger to smaller

cloud distance.
        Figure 5 presents contour maps of counts of XCO2raw-TCCON and XCO2bc-
TCCON over the ocean versus the CSNoiseRatio metric. As mentioned above, the
CSNoiseRatio values increase as the radiance field inhomogeneity (and cloudiness)
increases. The QF=0 data has most of the CSNoiseRatio values near unity, consistent with

spatially uniform radiance conditions. A wider range of CSNoiseRatio values is seen in the
QF=1 data, indicating relatively more observations impacted by spatially variable radiance.
The H(3D) and H(Continuum) variables have contour maps similar in visual appearance
to the Fig. 5 CSNoiseRatio contour map.
        Table 4 presents the minimum standard deviations in the data displayed in Fig. 4 and

5, and the range in the ratios of the standard deviations. Standard deviations in XCO2-
TCCON are calculated as a function of Distkm in bins of 2 km cloud distance for both
XCO2raw and XCO2bc. The minimum standard deviation is the smallest of the set of
standard deviations. The range of the standard deviations is the ratio of the largest to
smallest standard deviation in the set of standard deviations.  As an example, the ocean

QF=0 minimum standard deviations are 1.04 and 0.76 ppm for XCO2raw and XCO2bc in





Fig. 4 for the Distkm metric, while the ratios of maximum to minimum standard deviations are 1.16 and 1.26 for the XCO2raw and XCO2bc data. Table 4 also presents the minimum and standard deviation ratios for the H(3D), CSNoiseRatio, and H(Continuum) metrics. Generally, the minimum standard deviations are larger for the QF=1 case, the biased corrected standard deviations are less than the raw retrieval standard deviations, the ratios deviate from unity, and all metrics display these characteristics. If the OCO-2 retrievals were not susceptible to 3D cloud effects, then the ratios in the lower half of Table 4 would be close to unity, but this is not the case.


Further insight into the Fig. 4 and 5 distributions is presented in Fig. 6 and 7, in which averages and 95 % (2σ) confidence limits of the averages are displayed. The XCO2raw-TCCON and XCO2bc-TCCON averages become more negative for both QF=0 and QF=1 cases as cloud distance approaches zero in Fig. 6. The averages become closer to each other as nearest cloud distance increases to large values. Ideally, the XCO2bc-TCCON differences should approach zero as the nearest cloud distance becomes very large. Since the 95 % confidence limits do not overlap for small cloud distances, the differences in the averages, and the increasingly negative trend in the averages as cloud distance approaches zero, are statistically significant. This indicates that the operational bias correction does not completely remove 3D cloud effects from the XCO2raw retrievals for the full range of cloud distance. The operational bias correction makes the bias close to zero for the most frequent scenes, those that are close to clouds. The less-frequent far-from-clouds scenes end up with a +0.4 ppm bias because the bias correction scheme cannot get rid of the 3D cloud dependence.




The ocean *3D cloud bias* in Fig. 6 for QF=0 XCO2bc is near -0.4 ppm (the difference of 0 ppm at cloud distances near 0 km and 0.4 ppm at cloud distances greater than 10 km). For ocean QF=1 XCO2bc the 3D cloud bias is -2.2 ppm. Since 40 % (75 %) of the QF=0 (QF=1) data points observations over the ocean are within 4 km of clouds, it is apparent that many OCO-2 data points are subject to a negative 3D cloud bias that is not completely removed by the operational bias correction. The corresponding 3D cloud biases for XCO2bc-TCCON over the ocean for QF=0 and QF=1 data (for the CSNoiseRatio metric) are -1.3 and -1.4 ppm (see Fig. 7). The -1.4 ppm values is equal to the difference of -1.8 ppm (at the CSNoiseRatio of 7) minus -0.4 (at the CSNoiseRatio of 1). As mentioned above, radiance field inhomogeneity increases as the CSNoiseRatio increases. The XCO2bc-TCCON cloud biases for the QF=1 data for the Distkm and CSNoiseRatio variables, -2.2 and -1.4 ppm, differ somewhat in absolute size, but are consistent in sign (both are substantially negative).




Table 5 summarizes the 3D cloud biases derived from the four 3D metrics. In general, the cloud biases are all negative for the Distkm, CSNoiseRatio, and H(Continuum) 3D metrics over the ocean for the QF=0 data. The graph of the QF=1 XCO2bc-TCCON averages as a function of the H(3D) metric has a minimum at H(3D) near 0.9, maxima at H(3D) near 0.1 and 1.3, and a range of XCO2bc-TCCON averages that span 1.6 ppm. Table 5 indicates this non-linear (quadratic) curve characteristic with the ± symbol. Since the bias correction equations in Section 3 are based upon linear equations, the extension of these equations with linear H(3D) metric terms (see Section 11) is expected to be of limited utility.


The Table 5 cloud biases for V9 and V10 data are fairly close to each other. As an example, the V9 and V10 cloud biases for the cloud distance variable are -2.5 and -2.2 ppm



for QF=1 ocean data. These similarities indicate that 3D cloud effects persist irrespective of data version.

It is instructive to examine graphs of x=cloud distance versus y=dPsco2 (over the ocean) and x=cloud distance versus y=dPfrac (over land). Fig. 8 presents the averages and the 95 % confidence limits of the averages. dPsco2 is fairly constant for large cloud distances for QF=0 data, then becomes increasingly negative as cloud distance approaches zero. The range of dPsco2 is -0.6 and -3.6 hPa for the QF=0 and QF=1 ocean data, and the range of dPfrac is -0.3 and -2.2 ppm for the QF=0 and QF=1 land data. With 40 % and

75 % of the observations at distances less than 4 km for QF=0 and QF=1 data, the dependence of x=cloud distance and y= dPsco2 in Fig. 8 can be described by a linear line with positive slope (and less so for the y=dPfrac land data). Since dPsco2 and dPfrac are included in the operational bias correction (Eqns. (1) through (5) of Section 3), and these metrics are correlated to the cloud distance metric, the operational bias correction *indirectly*

"takes into account" 3D cloud effects.

## 8 Illustrative ocean scenes

While the previous section discussed global analyses, it is important to point out that 3D

cloud biases are readily apparent at local scales. Figure 9 displays glint data over the Pacific on June 12, 2016. MODIS clouds are indicated by irregular red shapes, while OCO-2 observations are indicated by color coded asterisks. For each horizontal row of asterisks there are eight OCO-2 footprints. Nearest cloud distance is indicated in the top panel, and H(Continuum) values are indicated in the middle panel. The H(Continuum) values increase

in size for the region surrounding the cloud at 15.6° N, with blue asterisks (low H(Continuum)) morphing into red and green asterisks (high H(Continuum)) as cloud distance decreases. In the bottom panel the Quality Flag becomes QF=1 for data points adjacent to this cloud feature.

    The upper panel of Fig. 10 presents XCO2bc versus nearest cloud distance from data

on June 12, 2016 for the 11° N – 17 °N, 158 ° E – 177° E range of latitude and longitude, which is larger than the Fig. 9 geospatial range. Only XCO2bc is graphed in Fig. 10 since TCCON data is not available for this ocean scene. At largest cloud distances the QF=1 XCO2bc data points span a limited range of XCO2bc, from 403 to 406 ppm. For the 0 to 2 km cloud distance range, the XCO2bc data points vary from 398 to 410 ppm, with a

noticeable "negative tail" of XCO2bc less than 403 ppm. Ranges of XCO2bc are binned into High, Mid, and Low bins of XCO2bc.

    The bottom panel of Fig. 10 presents average $O_2$ A-band spectra for the spectra associated with the three XCO2bc bins. The bottom panel indicates that 3D cloud effects perturb the "Mid" radiances in the $O_2$ A-band by ± 15 % in this glint scene. In a comparative

manner, the radiance perturbations for the $O_2$ A-band, WCO2, and SCO2 bands are ± (6, 7, 7) % and ± (15, 15, 18) % for the QF=0 and QF=1 cases. 3D cloud effect radiance perturbations are therefore large for all three bands.

    The operational retrieval iteratively solves for a state vector (which includes surface pressure, aerosol, surface reflectance, the $CO_2$ vertical profile, and other variables) that

matches observed and forward model radiances. Since 3D cloud radiative perturbations are not incorporated into the operational retrieval, the retrieved surface pressure, aerosol, surface reflectance, and $CO_2$ vertical profile, will differ from the actual atmospheric values.

These differences will increase as the severity of the 3D cloud effect increases at small cloud distances. Since 3D cloud effects perturb all bands, the retrieved surface pressure
differs from the actual surface pressure, and this difference propagates into the XCO2raw retrieval.

For a range of latitude (52° S - 41°S) and longitude (164° E - 180° E), with Lauder, New Zealand being the closest TCCON site, Fig. 11 displays scatter diagrams of TCCON – XCO2bc, CSNoiseRatio, dPsco2, CO2graddel, DWS, and $O_2$ A-band surface reflectance,
as a function of cloud distance. All observations during 2017, for which TCCON data is matched to the OCO-2 observations, are considered, with most of the data points observed during November and February. The QF=0 and QF=1 data points in Fig. 11 are color coded by green and red symbols, respectively. The various panels consistently indicate that dPsco2 and CO2graddel values are near zero for QF=0 data points, and are accompanied
by low DWS, surface reflectance, and CSNoiseRatio values, for both small and large cloud distances. The measured QF=1 CSNoiseRatio becomes progressively larger as cloud distance decreases. For QF=1 data the dPsco2, CO2graddel, DWS, and surface reflectance variables take on unrealistic values as cloud distance decreases from large to small values.. These unrealistic values are necessary in order for the retrieval to match observed and
forward model radiances. When the 3D cloud effect adds radiance to the observations, a large DWS or reflectance value is able to increase the forward model radiance to the measured radiance.

**9 XCO2 Cloud Bias Mitigation by Table look-up correction factors**

Figures 6 and 7 suggest mitigation of 3D cloud biases by application of a Table look-up correction. Using the CSNoiseRatio QF=1 data as an example, and the XCO2raw data points, for a given XCO2raw data point there is a corresponding CSNoiseRatio value and XCO2raw-TCCON average (see the upper right panel in Fig. 7). The corrected XCO2raw
value (XCO2raw,corr) is then simply the XCO2raw value *minus* the XCO2raw-TCCON average. The lower right panel of Fig. 7 can be used in a similar calculation to specify QF=1 XCO2bc,corr values. Note that these Table look-up table mitigation calculations can be applied after the operational bias-correction processing, with XCO2raw,corr and XCO2bc,corr data added to the data included in Lite files, provided that the CSNoiseRatio
and/or Distkm values that correspond to the OCO-2 observations are known.

Table 6 presents statistics of Table look-up cloud bias mitigation calculations, corresponding to calculations in which the four 3D metrics are applied separately to the raw and bc data. The two "Standard" rows in Table 6 refer to standard deviations and PDF averages of XCO2bc-TCCON, based upon Lite file XCO2bc. The rest of Table 6 then
presents the statistics (PDF averages and standard deviations of XCO2raw,corr-TCCON and XCO2bc,corr - TCCON) of the ocean and land QF=0 and QF=1 corrected data, for the four 3D metrics.

Table 6 indicates that the Table look-up technique changes XCO2-TCCON averages, but not their standard deviations. The XCO2bc,corr-TCCON standard deviations for QF=0
and QF=1 data over land and ocean are close to the standard deviations of the "Standard" values. The "Standard" XCO2bc-TCCON averages for QF=1 ocean and land data are near -1 ppm, while the corrected XCO2bc,corr data has PDF averages near or less than 0.2 ppm, depending upon which 3D metric (and its associated set of XCO2bc-TCCON averages) is





applied. Since the XCO2bc-TCCON "Standard" averages are already small (0.3 ppm and 0.11 for QF=0 data over ocean and land), the Table look-up mitigation technique is therefore more beneficial for the QF=1 XCO2bc data than for the QF=0 XCO2bc data.

The data in Table 6, however, does not reveal a shortcoming of the Table look-up mitigation technique, when only a *single* 3D metric is applied. Using the CSNoiseRatio 3D metric as an example, the use of the Fig. 7 CSNoiseRatio averages yields a corrected set of XCO2bc,corr values and new XCO2bc,corr – TCCON averages (in a revised Fig. 7 graph, not shown) in which the new averages are very close to zero, binned as a function of CSNoiseRatio. The corresponding revised Fig. 6, based upon the CSNoiseRatio correction, however, displays a large range of XCO2bc,corr – TCCON averages when the averages are binned as a function of Distkm.

The general situation is indicated in Fig. 12. The x and y axes are bins of Distkm and CSNoiseRatio, with contouring of XCO2raw – TCCON and XCO2bc – TCCON from -5 to 1 ppm. In the construction of Fig. 12, the adopted Distkm and CSNoiseRatio set of bins had a finer (coarser) bin increment for small (large) values of Distkm and CSNoiseRatio, in order to include a similar number of data points for each x-y grid cell. In Fig. 12 the largest variation in XCO2raw – TCCON and XCO2bc – TCCON is present along the Distkm axis, especially for the QF=1 data, while the variation is smaller along the CSNoiseRatio axis (e.g. for small Distkm values). Though the Table 6 CSNoiseRatio "bc ave" value of XCO2bc,corr – TCCON for QF=0 (QF=1) ocean data is near 0.06 (0.09) ppm, the revised Fig. 6 graph indicates that the XCO2bc,corr – TCCON averages vary by 0.3 (-1.9) ppm as a function of the Distkm metric. The mitigation of the cloud bias by the CSNoiseRatio 3D metric therefore does not remove the 3D cloud bias when one examines the 3D cloud bias in a XCO2bc,corr – TCCON versus Distkm graph.

Using the Fig. 12 data as the basis for a Table look-up correction, new Fig. 6 and 7 averages are displayed in Fig. 13 and 14, and were calculated as follows. For a given pair of Distkm and CSNoiseRatio values that are associated with a single XCO2 measurement, the Fig. 12 XCO2raw – TCCON or XCO2bc – TCCON values for the specific Distkm, CSNoiseRatio pair is subtracted from the XCO2raw and XCO2bc values. Applying the Fig. 12 corrections to all of the XCO2 measurements, Fig. 13 and 14 indicate that the revised XCO2raw,corr – TCCON and XCO2bc,corr – TCCON averages are then within ± 0.2 ppm of zero for both 3D metrics. Figures (not shown) for the corresponding corrected averages over land are also within ± 0.2 ppm of zero, with the exception of one data point. The utilization of the Fig. 12 data, in which both Distkm and Cloudratio 3D metrics are used in a Table look-up application, appears to be a better way to mitigate for 3D cloud biases compared to single-variable Table look-up calculations.

An additional calculation was carried out in which the Fig. 12 data was fit by linear regression, represented by a constant term plus Distkm and CSNoiseRatio terms. Four x-y fits were calculated, one for each of the four panels in Fig. 12. This representation was then applied as the basis for correction of the XCO2 data. This calculation yielded Fig. 13 and 14 style graphs which had *larger ranges* in the XCO2raw,corr – TCCON and XCO2bc,corr – TCCON averages than those based upon the Fig 12. Table look-up technique.

Figure 12 therefore has variations which are not easy to represent by a linear regression. This has bearing upon the calculations discussed below in Section 11 in which 3D metrics are added to the operational bias correction equations. The comparison here of the two calculations, based upon the Table look-up and x-y linear regression representations of the

Fig. 12 data, suggests that the Table look-up technique is a better 3D cloud bias mitigation technique.

## 10 Mitigation by data screening

Another way to mitigate for 3D cloud biases is to apply 3D metric *data screening*. Table 7 presents standard deviations and PDF averages of XCO2bc-TCCON over the ocean for various data screening thresholds, and is read in the following manner. Referring to Distkm as the nearest cloud distance, ocean QF=0 XCO2bc-TCCON data for Distkm between 2 and 50 km has a standard deviation of 0.80 ppm, with a sample size fraction of 0.83 of the

total possible number of QF=0 data points, and the average of the XCO2bc-TCCON PDF is 0.36 ppm. For Distkm between 5 and 50 km, the standard deviation is 0.78, with a sample fraction of 0.62 of the QF=0 data points, and the PDF average is 0.40 ppm. For QF=1 data the standard deviations for these two Distkm screening thresholds are 2.03 and 1.89 ppm, with sample fractions of 0.41 and 0.19, with PDF averages of -0.16 to 0.36 ppm.

660       Table 7 indicates that the PDF averages are already acceptable for QF=0 ocean data, since PDF averages (in absolute value) are less than 0.5 ppm (a reasonable mitigation goal) when no screening is done. For QF=1 ocean data, however, the standard deviations and PDF averages change substantially as the cloud distance threshold screening is applied. If all data points are accepted, then the standard deviation is near 2.3 ppm, and the XCO2bc-

TCCON PDF average is near -0.99 ppm. For a cloud distance threshold near 1 km the data screening reduces the average of the XCO2bc – TCCON PDF to near 0.5 ppm (in absolute value), with a sample fraction near 0.60.

        H(3D), CSNoiseRatio, and H(Continuum) screening thresholds, and their associated standard deviations and XCO2bc-TCCON PDF averages over the ocean are also

summarized in Table 7. For the QF=0 data the data screening changes the deviations and averages by very small amounts. For the QF=1 data the data screening yields substantial changes in the deviations and PDF averages. The H(3D), H(Continuum), and CSNoiseRatio, screening thresholds of 0.57, 14, and 4.2 yield XCO2bc-TCCON PDF averages (in absolute value) near 0.5 ppm, with sample fractions of 0.72, 0.73, and 0.70.

We note that the H(Continuum) and CSNoiseRatio metrics, however, are from *stand-alone* OCO-2 measurements, while the nearest cloud distance and H(3D) metrics rely upon MODIS measurements.

        Table 8 indicates that the PDF averages are already acceptable for QF=0 land data, since PDF averages (in absolute value) are less than 0.5 ppm when no screening is done

For QF=0 data, with no data screening, the standard deviations over land (near 1.2) are larger than those over the ocean (near 0.8, see Table 7). For QF=1 data, the changes are substantial, with deviations changing from 4 to 2 ppm for the Distkm screening, and from 3.6 to 2.8 ppm for the other metrics. The PDF averages decrease to the 0.5 ppm level (in absolute value) when approximately 0.65 of the Distkm data points are utilized, by only

using data with nearest cloud distances greater than 2.2 km. While the CSNoiseRatio metrics do not decrease the XCO2bc-TCCON deviations and PDF averages to the 0.50 ppm level (see column 12 of Table 8), the PDF averages decrease to the 0.8 ppm level (in absolute value) when approximately 0.63 of the CSNoiseRatio data points are utilized, by only using data with CSNoiseRatio values less than 3.4.





Figure 15 displays the changes in the PDFs over the ocean and land as a function of nearest cloud distance screening thresholds. The PDFs correspond to the data summarized in Tables 6 and 7. Generally, the PDFs change very little for the QF=0 data over ocean and land. The PDFs essentially lie atop each other. The largest changes are apparent over ocean and land for the QF=1 data. The data screening reduces the negative XCO2bc-TCCON

"tail" data points. These "tail" data points are apparent in Fig. 4, 5, 10, and 11.

Graphs (not shown) of the PDFs for CSNoiseRatio screening thresholds, and thresholds for the H(3D) and H(Continuum) metrics, have a visual appearance similar to the Fig. 15 graphs. The QF-=0 PDFs lie atop each other, while the QF=1 data screening reduces the negative XCO2bc-TCCON "tail" data points.

One concludes from Tables 7 and 8 and Fig. 15 that it is possible to screen the QF=1 XCO2bc data using the Distkm or CSNoiseRatio 3D metrics to improve the standard deviations of XCO2bc-TCCON, and to reduce the XCO2bc-TCCON PDF averages to the 0.5 ppm level for the ocean data, yet this is done by a screening process which tosses out approximately 30 to 40 % of the converged retrieval QF=1 data points. For the land data

the 0.5 (0.8) PDF average absolute value occurs in Distkm (CSNoiseRatio) data screening when 35 % of the data points are excluded. None of the screenings change the QF=1 standard deviations to those approaching the 0.8 ppm and 1.2 ppm standard deviations of the ocean and land QF=0 data.

**11 Mitigation by additional linear-regression terms**

The possibility of mitigating 3D cloud biases by adding terms to the bias correction process, was investigated by adding one or more 3D metrics to Eqns. (1)-(3). Each application of the Interactive Data Language (IDL) Regress linear regression routine solved

for new Eqns. (2) and (3) linear coefficients, and new XCO2bc-TCCON standard deviations and PDF averages.

Table 9 presents representative comparisons of the two sets of calculations. Available data points, for which Distkm values were well determined for 60° S to 60° N, were used in the generation of Table 9. Two vertically adjacent numbers are tabulated for the QF=1

data. The top number is the value calculated when all possible data points are included in the regressions, while for the bottom entry the ranges of dPsco2 and CO2graddel (for oecan) and dPfrac, CO2graddel, and logDWS (for land) are equal to those ranges for the QF=0 data. The QF=0 (best quality) data points follow from the operational methodology of limiting dPsco2, DPfrac, CO2graddel (and other variables) to narrow limited ranges (see

Version 9 OCO-2 Data Product User's Guide (2018) for a discussion of these ranges), The two vertically adjacent entries therefore indicate the sensitivity of the XCO2bc-TCCON XCO2 PDF standard deviations to the dPsco2, DPfrac, CO2graddel range limits.

The number of data points for the regression, the standard deviation of the XCO2bc-TCCON differences (based upon the new set of regression coefficients), and also an

additional "maxlatDiff" metric are tabulated. PDF XCO2bc-TCCON averages are not presented in Table 9 since they are close to zero for all regression calculations. The "latmaxDiff" metric is calculated by first calculating XCO2bc-TCCON averages for 20° latitude bands from 60° S to 60° N, and then calculating maxlatDiff as the difference in the maximum and minimum of the five averages. If the bias correction is accurate globally,

then the XCO2bc-TCCON averages should have little latitudinal variation. If this is not the





case, then the latitudinal gradients associated with bias correction introduce XCO2bc latitudinal gradients (large maxldatDiff values) that will be problematic for those using OCO-2 XCO2bc to infer regional $CO_2$ vertical fluxes in "flux inversion" modeling studies.

Adding Distkm, H(3D), CSNoiseRatio, and H(Continuum) variables individually to the linear regressions does not significantly produce smaller XCO2bc-TCCON standard deviations or smaller maxlatDiff values, compared to the regressions that do not include these additional terms. The largest differences in Table 9 are due to imposing narrow ranges of dPsco2, DPfrac, and CO2graddel for the QF=1 data.

Graphs of alog (Cloud Distance) versus XCO2bc-TCCON averages (not shown) are linear, while the Fig. 6 graphs of Cloud Distance versus XCO2bc-TCCON averages are not, suggesting that it is useful to add alog (Distkm), instead of Distkm to the linear regression. The application of alog (Distkm) in the linear regression equations changed the standard deviations slightly for the QF=0 ocean data. Repeating the linear regressions for various terms, e.g. exp(-Distkm/5.0), alog(CSNoiseRatio), in separate calculations, or the

addition of two terms (alog(Distkm) and sun-cloud-observation footprint scattering angle) or three terms (alog(Distkm), alog(CSNoiseRatio), alog(H(3D))), only yielded marginal decreases in the XCO2bc-TCCON standard deviations. Extending Eqns. (2) and (3) to include quadratic terms, e.g. Eqn. (2) with dPsco2, dpSCO2$^2$, CO2graddel, CO2graddel$^2$, Distkm, and Distkm$^2$ terms, only improved marginally the XCO2bc-TCCON standard

deviations.

## 12 Discussion

Overall, the OCO-2 cloud pre-processor is effective in identifying clouds, but observations

impacted by low altitude clouds and 3D scattering effects are sometimes not identified. The Lite files contain many observations that are close to clouds, with 40 % and 75 % of OCO-2 Lite file retrievals (see Table 2) within 4 km of clouds over the ocean and land for the QF=0 and QF=1 cases (Fig. 1). 3D radiative transfer calculations for the same cloud field (with representative surface reflectance over the ocean and land, for ocean glint and

land nadir viewing geometry) indicate that ocean 3D cloud radiance perturbations are larger over the ocean than over land (Fig. 2) at this cloud distance.

There is a marked contrast in the Lite file QF=0 and QF=1 OCO-2 data. Figures 1 and 4 indicate that QF=1 data points are closer to clouds on average than the QF=0 data points. Figure 4 visually indicates that there is a strong asymmetry in XCO2bc-TCCON, with more

negative values than positive values for a small nearest cloud distances. Though both sets of measurements reached convergence in the operational retrieval, only the QF=0 data points are used in operational post-retrieval bias correction calculations.

From a pragmatic perspective, it is important to consider a variety of 3D cloud metrics, since the Distkm and H(3D) metrics require the processing of auxiliary MODIS cloud and

radiance fields. The CSNoiseRatio and H(Continuum) metrics are calculated from *stand-alone* OCO-2 measurements. Furthermore, OCO-2 views the Earth's surface six minutes before MODIS Aqua, so some clouds observed by MODIS may not be present when OCO-2 makes observations. The Distkm metric is a cloud field metric, while the H(3D), CSNoiseRatio, and H(Continuum) metrics are measures of radiance field inhomogeneity.

Surface reflectivity variations, variations not related to 3D cloud radiative effects, contribute to all three of these radiance field metrics.





Figures 6 and 7 indicate that the Version 10 bias-corrected retrievals have a non-zero residual 3D cloud bias. The XCO2bc-TCCON averages become more negative as the nearest cloud distance decreases, and as the CSNoiseRatio increases. From Table 5, XCO2bc –TCCON values at small cloud distances differ from those at large cloud distances by -0.4 and -2.2 ppm for the QF=0 and QF=1 data over the ocean.

While the previous discussion pertains to global statistics, 3D cloud effects are readily apparent at local scales of several degrees of longitude and latitude. This is illustrated by Fig. 9, in which nearest cloud distance, H(Continuum), and Quality Flag data is presented on a footprint by footprint basis. QF=1 and larger H(Continuum) values are located right next to clouds. Figure 10 presents XCO2bc as a function of nearest cloud distance for a larger region containing the local region presented in Fig. 9. The asymmetry in XCO2bc is readily apparent in Fig. 10, consistent with the asymmetry present in Fig. 4. The bottom panel of Fig. 10 illustrates for QF=1 spectra that there is a ± 15 % variation in radiance, compared to the "Mid" radiance values, in the $O_2$ A-band for this scene. 3D cloud radiative perturbations are large for all three OCO-2 spectral bands.

The operational retrieval iteratively solves for a state vector (which includes surface pressure, aerosol, surface reflectance, the $CO_2$ vertical profile, and other variables) that matches observed and forward model radiances. Since 3D cloud effect perturbations, illustrated in Fig. 10, are not incorporated into the operational retrieval, the surface pressure, aerosol, surface reflectance, and $CO_2$ vertical profile, will differ from the actual atmospheric values. These differences increase as the severity of the 3D cloud effect increases at small cloud distances. This is apparent in Fig. 11 in which ocean bias correction (dPsco2, CO2graddel), land bias correction (DWS, and CO2graddel), and other variables (surface reflectance, and CSNoiseRatio) increase as the nearest cloud distance decreases for the QF=1 data. These variables have a much larger range in value than for the QF=0 data.

Figure 15 displays XCO2bc-TCCON PDFs calculated for a set of nearest cloud thresholds from 0 to 15 km. A 5 km threshold means that only XCO2bc data with nearest cloud distances *greater* than 5km are utilized. For the QF=0 data the PDFs essentially lie atop each other. *Data screening* (see Tables 6 and 7) does not reduce the XCO2bc-TCCON averages for QF=0 data, since they are low (less than 0.5 ppm in absolute value, for ocean and land data) for data populations which include all observations. For the QF=1 data, the PDFs have negative XCO2bc-TCCON tails. Tables 7 and 8 indicate that the QF=1 3D cloud biases can be reduced to the 0.5 ppm level over the ocean if approximately 60 % (70 %) of the QF=1 data points are utilized, by applying Distkm (CSNoiseRatio) metrics in a data screening process. Over land the QF=1 3D cloud biases can be reduced to the 0.5 ppm level if approximately 65 % of the QF=1 data points are utilized, by data screening based upon the Distkm metric, and to the 0.8 ppm level if 63 % of the QF=1 data points are utilized based upon CSNoiseRatio data screening.

Table 9 indicates that adding additional variables to the current multi-variable linear regression bias correction, without data screening, will only improve XCO2bc-TCCON standard deviations by a slight amount. The four 3D cloud metrics (Distkm, H(3D), CSNoiseRatio, and H(Continuum)) were added to the present bias equations individually in separate linear regression calculations. XCO2bc-TCCON standard deviations only improved in the second decimal place. Other terms, such as alog(Distkm), or combination



of terms (alog(Distkm), alog(CSNoiseRatio), alog(H3D)) also did not yield a dramatic improvement in the statistics.

Comparing the three mitigation techniques: a) Table look-up (Section 9), b) data screening (Section 10), and c) linear-regression (Section 11), adding terms to the linear-regression equations had the least beneficial improvement in XCO2bc-TCCON statistics. The Table look-up and data screening techniques both are able to reduce XCO2bc-TCCON QF=1 averages to the 0.5 ppm level. The Table look-up technique that uses two 3D metrics (Distkm and CSNoiseRatio, see Fig. 12) provides the best reduction in 3D cloud bias. One

advantage of the Table look-up technique, compared to the data screening technique, is that data points are not thrown out from localized scenes. This is especially useful for regions in the tropics that have relatively few OCO-2 retrievals. Table look-up (Figures 6, 7 and 12) and 3D metrics (Distkm, H(3D), H(Contimuum), CSNoiseRatio for Lite file observations) will be placed in publically available data files. These data files can be used

in application of the techniques discussed in this paper (or by other user-developed techniques) to mitigate the 3D cloud effects that are present in OCO-2 XCO2 data.

*Data availability*. The TCCON data can be obtained from the TCCON Data Archive hosted by CaltechDATA at https://tccondata.org. The 3D metrics (based upon Version 9

and 10 data), corresponding to Lite file observations, and associated data (such as Figures 6, 7 and 12, which apply to version 10 OCO-2 data), can be downloaded from the CERN based Zenodo archive (https://doi.org/10.5281/zenodo.4008765).

*Author Contribution*. Steven Massie performed many of the calculations presented in this

paper, and was the primary author of the text. Heather Cronk created the CSU MODIS files. Aronne Merrelli created the colorslice derived metrics and produced the merged data sets that combined the OCO-2 XCO2, TCCON, and 3D metrics into convenient single files. Sebastian Schmidt and Hong Chen provided suggestions on the content of the paper. David Baker provided suggested modifications and clarifications in the text.


*Competing interests*. The authors declare that they have no conflict of interest.

*Acknowledgements*. STM, KSS, and HC acknowledges support by NASA Grant 80NSSC18K0889 "Towards Detection and Mitigation of 3D Cloud Effects and XCO2

Retrievals". AM acknowledges support by NASA Grants NNX15AH96G and 80NSSC18K0891. Appreciation is expressed to Chris O'Dell for assistance with TCCON data matchup and processing of early OCO-2 version 10 test data. Appreciation is expressed to the TCCON teams who measure and provide ground based XCO2 validation to the carbon cycle research community. Appreciation is expressed to the OCO-2 computer

staff at the Jet Propulsion Laboratory, and to Garth D'Attillo and Timothy Fredrick of the Atmospheric Chemistry Observations and Modeling (ACOM) division at the National Center for Atmospheric Research (NCAR), supported by the National Science Foundation, for maintaining the operational capabilities of computer systems during 2020, a challenging year due to the ongoing global COVID-19 pandemic.






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





Table 1. List of TCCON sites and their locations.

|      | Site | Latitude | Longitude | Reference |
|------|------|----------|-----------|-----------|
| 1085 | Anmyeondo, Korea | 36.53 | 126.33 | Goo et al. (2014) |
|      | Armstrong, USA | 34.59 | -117.88 | Iraci et al. (2016) |
|      | Bialystok, Poland | 53.23 | 23.02 | Deutscher et al. (2015) |
|      | Bremen, Germany | 53.10 | 8.85 | Notholt et al. (2014) |
|      | Borgos, Philippines | 18.53 | 120.65 | Velazco et al. (2017) |
| 1090 | Caltech, USA | 34.13 | -118.12 | Wennberg et al. (2015) |
|      | East Trout Lake, Canada | 54.35 | -104.98 | Wunch et al. (2016) |
|      | Garmisch, Germany | 47.47 | 11.06 | Sussmann, Rettinger (2014) |
|      | Izana, Tenerife | 28.3 | -16.5 | Blumenstock et al. (2014) |
|      | Karlsruhe, Germany | 49.10 | 8.43 | Hase et al. (2015) |
| 1095 | Lamont, OK, USA | 36.60 | -97.48 | Wennberg et al. (2016) |
|      | Lauder, New Zealand | -45.03 | 169.68 | Sherlock et al. (2014) |
|      | Orleans, France | 47.97 | 2.11 | Warneke et al. (2014) |
|      | Paris, France | 48.84 | 2.35 | Te et al. (2014) |
|      | Park Falls, WI, USA | 45.94 | -90.27 | Wennberg et al. (2014) |
| 1100 | Réunion Island | -20.90 | 55.48 | De Mazière et al. (2014) |
|      | Rikubetsu, Japan | 43.45 | 143.76 | Morino et al. (2016b) |
|      | Saga, Japan | 33.24 | 130.28 | Kawakami et al. (2014) |
|      | Sodankyla, Finland | 67.36 | 26.63 | Kivi and Heikkinen (2016) |
|      | Tsukuba, Japan | 36.05 | 140.12 | Morino et al. (2016a) |
| 1105 | Wollongong, Australia | -34.40 | 150.87 | Griffith et al. (2014) |



Table 2. The fractions of OCO-2 Lite file observations (in percent) that have a cloud within 4 km of an observation footprint for each season.[a]

| Season | Ocean, QF=0 | Land, QF=0 | Ocean, QF=1 | Land, QF=1 |
|---|---|---|---|---|
| Winter | 30-54 | 30-53 | 61-90 | 61-96 |
| Average | 37 | 42 | 79 | 77 |
| Spring | 32-55 | 31-53 | 73-88 | 60-83 |
| Average | 42 | 42 | 80 | 73 |
| Summer | 30-57 | 29-56 | 59-89 | 58-82 |
| Average | 41 | 39 | 79 | 70 |
| Fall | 21-58 | 24-55 | 55-88 | 59-83 |
| Average | 41 | 38 | 78 | 70 |

[a]The two tabulated numbers are the minimum and maximum values of the fractions (in %) for five 20º latitudinal bins (see Fig. 1). The Average value is the average of the fractions of the latitudinal bins.




Table 3. Input to SHDOM calculations.[a]

| Variable | Base State | Perturbation |
|---|---|---|
| Surface Pressure (hPa) | 1016 | 1026 |
| Surface Reflectance (nadir) | 0.32, 0.21, 0.11 | 0.35, 0.23, 0.12 |
| Wind velocity (glint) | 10, 10, 10 m sec$^{-1}$ | 15, 15, 15 m sec$^{-1}$ |
| Aerosol Optical Depth | 0,11, 0.06, 0.048 | 0.165, 0.09, 0.072 |
| $CO_2$ (ppm) | 400 | 410 |




[a]The triplet of numbers refer to the O2, WCO2, and SCO2 bands, respectively.
Perturbations are applied individually one at a time, keeping all other variables to their
base state values.





Table 4. Minimum standard deviations (ppm) and ranges of the ratios of the Version 10 XCO2-TCCON standard deviations.[a]


| | Minimum standard deviations | | | |
|---|---|---|---|---|
| Metric | Ocean, QF=0 | Land, QF=0 | Ocean, QF=1 | Land, QF=1 |
| Cloud Distance | 1.04 (raw) | 1.75 | 1.64 | 2.79 |
| | 0.76 (bc) | 1.20 | 1.45 | 2.18 |
| H(3D) | 0.98 | 1.62 | 1.95 | 2.57 |
| | 0.69 | 1.03 | 1.91 | 1.73 |
| CSNoiseRatio | 1.04 | 1.68 | 2.02 | 2.69 |
| | 0.79 | 1.11 | 1.78 | 2.28 |
| H(Continuum) | 0.98 | 1.45 | 1.74 | 1.91 |
| | 0.72 | 0.96 | 1.18 | 1.97 |




| | Ranges of the standard deviation ratios[b] | | | |
|---|---|---|---|---|
| Metric | Ocean, QF=0 | Land, QF=0 | Ocean, QF=1 | Land, QF=1 |
| Cloud Distance | 1.16 (raw) | 1.14 | 1.41 | 1.26 |
| | 1.26 (bc) | 1.19 | 1.62 | 1.70 |
| H(3D) | 1.20 | 1.79 | 1.20 | 1.45 |
| | 1.43 | 1.70 | 1.23 | 2.08 |
| CSNoiseRatio | 1.22 | 1.14 | 1.25 | 1.37 |
| | 1.74 | 1.11 | 1.52 | 1.51 |
| H(Continuum) | 1.36 | 1.52 | 1.55 | 2.00 |
| | 1.43 | 1.53 | 2.36 | 1.70 |





[a]The pairs of numbers refer to raw and bias corrected (bc) XCO2.
[b]The range of the standard deviation ratios is the maximum standard deviation divided by the minimum standard deviation of the set of standard deviations for a given metric, surface type, and QF flag.






Table 5. 3D cloud biases for bias corrected V9 and V10 XCO2.[a]

| Metric | Ocean, QF=0 | Ocean, QF=1 | Land, QF=0 | Land, QF=1 |
|--------|-------------|-------------|------------|------------|
| Cloud Distance | -0.5 (V9) | -2.5 | 0.05 | -3.3 |
|  | -0.4 (V10) | -2.2 | ±0.1 | -2.5 |
| H(3D) | ±0.5 | ±1.6 | ±1 | ±2 |
|  | ±0.3 | ±2.0 | 0.4 | ±2.2 |
| CSNoiseRatio | -1.5 | -1.9 | 0.3 | -1 |
|  | -1.3 | -1.4 | 0.15 | -0.9 |
| H(Continuum) | -0.8 | -2.0 | 0.5 | ±5 |
|  | -0.4 | -1.5 | 0.5 | ±3.7 |

[a] There are two paired numbers. The top number is for Version 9 data, while the bottom number is for Version 10 data. A negative 3D cloud bias indicates that XCO2bc is less than TCCON XCO2. A ± value indicates that the graph of e.g. H(3D) versus XCO2bc – TCCON is not monotonic (i.e. there is a maximum or minimum of the graph in the middle of the graph). The cloud biases are read off from inspection of Fig. 6 and 7 (i.e. the range in y axis values) and corresponding graphs of x=H(3D), CSNoiseRatio or H(Continuum) versus y= XCO2bc – TCCON in other graphs (not shown).





Table 6. Statistics of the single variable Table look-up cloud bias mitigation calculations.[a]

| Metric | | Ocean, QF=0 | Ocean, QF=1 | Land, QF=0 | Land, QF=1 |
|---|---|---|---|---|---|
| Standard | bc stnd | 0.83 | 2.33 | 1.21 | 3.88 |
| | bc ave | 0.30 | -0.98 | 0.11 | -1.06 |
| Distkm | raw stnd | 1.09 | 2.32 | 1.80 | 3.64 |
| | bc stnd | 0.82 | 2.19 | 1.21 | 3.78 |
| | raw ave | 0.02 | 0.00 | 0.00 | 0.07 |
| | bc ave | 0.00 | 0.01 | -0.02 | 0.08 |
| H(3D) | raw stnd | 1.06 | 2.36 | 1.74 | 3.48 |
| | bc stnd | 0.80 | 2.21 | 1.15 | 3.56 |
| | raw ave | 0.09 | 0.12 | -0.21 | -0.18 |
| | bc ave | 0.02 | -0.04 | -0.11 | -0.06 |
| CSNoiseRatio | raw stnd | 1.06 | 2.39 | 1.74 | 3.54 |
| | bc stnd | 0.80 | 2.23 | 1.15 | 3.62 |
| | raw ave | 0.11 | 0.17 | -0.13 | 0.10 |
| | bc ave | 0.06 | 0.08 | -0.11 | 0.20 |
| H(Continuum) | raw stnd | 1.07 | 2.39 | 1.74 | 3.53 |
| | bc stnd | 0.81 | 2.26 | 1.15 | 3.62 |
| | raw ave | 0.03 | 0.13 | -0.11 | 0.00 |
| | bc ave | 0.00 | 0.03 | -0.09 | 0.22 |

[a]The first two "Standard" rows of the Table refer to the standard deviations (stnd, in ppm) and averages of XCO2bc –TCCON, with XCO2bc from the Lite files. The four rows for each metric report the standard deviations and averages of $XCO2_{raw,corr}$ – TCCON and $XCO2_{bc,corr}$ – TCCON.





Table 7. Standard Deviations (in ppm) of Version 10 XCO2bc-TCCON XCO2 over the ocean for various Distkm, H(3D), H(Continuum), and CSNoiseRatio thresholds [a]

1265

---

### Quality flag=0

| Range | | | | Standard Deviations | | | | PDF Average | | | | Fraction of Data Points | | | |
|---|---|---|---|---|---|---|---|---|---|---|---|---|---|---|---|
| 0 | 1.0 | 40 | 20 | 0.84 | 0.81 | 0.81 | 0.81 | 0.31 | 0.32 | 0.32 | 0.32 | 1.00 | 1.00 | 1.00 | 1.00 |
| 1 | 0.8 | 30 | 10 | 0.82 | 0.81 | 0.81 | 0.81 | 0.34 | 0.32 | 0.32 | 0.32 | 0.91 | 0.98 | 0.99 | 1.00 |
| 2 | 0.6 | 20 | 8 | 0.80 | 0.80 | 0.81 | 0.81 | 0.36 | 0.33 | 0.33 | 0.32 | 0.83 | 0.95 | 0.98 | 1.00 |
| 3 | 0.4 | 15 | 5 | 0.79 | 0.79 | 0.80 | 0.81 | 0.38 | 0.34 | 0.33 | 0.32 | 0.75 | 0.90 | 0.96 | 1.00 |
| 5 | 0.3 | 10 | 3 | 0.78 | 0.78 | 0.80 | 0.81 | 0.40 | 0.36 | 0.33 | 0.33 | 0.62 | 0.85 | 0.93 | 0.99 |
| 10 | 0.2 | 5 | 2 | 0.77 | 0.77 | 0.77 | 0.79 | 0.41 | 0.37 | 0.35 | 0.35 | 0.39 | 0.78 | 0.78 | 0.94 |
| 15 | 0.1 | 2 | 1 | 0.77 | 0.77 | 0.72 | 0.77 | 0.41 | 0.40 | 0.40 | 0.41 | 0.24 | 0.66 | 0.31 | 0.51 |

1270 / 1275

---

### Quality Flag =1

| Range | | | | Standard Deviations | | | | PDF Average | | | | Fraction of Data Points | | | |
|---|---|---|---|---|---|---|---|---|---|---|---|---|---|---|---|
| 0 | 1.0 | 40 | 20 | 2.34 | 2.33 | 2.22 | 2.33 | -0.99 | -0.84 | -0.72 | -0.86 | 1.00 | 1.00 | 1.00 | 1.00 |
| 1 | 0.8 | 30 | 10 | 2.12 | 2.31 | 2.17 | 2.24 | -0.51 | -0.75 | -0.67 | -0.79 | 0.60 | 0.91 | 0.95 | 0.96 |
| 2 | 0.6 | 20 | 8 | 2.03 | 2.25 | 2.05 | 2.19 | -0.16 | -0.54 | -0.58 | -0.74 | 0.41 | 0.75 | 0.85 | 0.92 |
| 3 | 0.4 | 15 | 5 | 1.96 | 2.09 | 1.96 | 2.07 | 0.09 | -0.21 | -0.52 | -0.58 | 0.30 | 0.53 | 0.76 | 0.79 |
| 5 | 0.3 | 10 | 3 | 1.89 | 1.95 | 1.81 | 1.94 | 0.36 | -0.01 | -0.43 | -0.38 | 0.19 | 0.41 | 0.60 | 0.58 |
| 10 | 0.2 | 5 | 2 | 1.86 | 1.82 | 1.56 | 1.83 | 0.54 | 0.22 | -0.22 | -0.21 | 0.11 | 0.31 | 0.30 | 0.40 |
| 15 | 0.1 | 2 | 1 | 1.80 | 1.61 | 1.33 | 1.51 | 0.53 | 0.42 | 0.22 | 0.18 | 0.06 | 0.21 | 0.05 | 0.12 |

1280 / 1285 / 1290

---

[a] Columns 1-4 refer to Distkm, H(3D), H(Continuum), and CSNoiseRatio data screening thresholds. In the first column, "2" indicates that Distkm data from 2 to 50 km are utilized, yielding a standard deviation for QF=0 data of 0.80 (column 5), with an average PDF XCO2(bc) – T CCON XCO2 of 0.36 ppm (column 9), with a fraction of 0.83 of the total number of data points being utilized (column 13).

1295





Table 8. Standard Deviations (in ppm) of Version 10 XCO2bc-TCCON XCO2 over land
for various Distkm, H(3D), H(Continuum), and CSNoiseRatio thresholds. [a]

1300 -------------------------------------------------------------------------------------------------------

Quality flag=0

-------------------

| | Range | | | Standard Deviations | | | | PDF Average | | | | Fraction of Data Points | | | |
|---|---|---|---|---|---|---|---|---|---|---|---|---|---|---|---|
| 0 | 1.0 | 40 | 20 | 1.22 | 1.14 | 1.14 | 1.15 | 0.12 | 0.01 | 0.00 | 0.00 | 1.00 | 1.00 | 1.00 | 1.00 |
| 1 | 0.8 | 30 | 10 | 1.22 | 1.14 | 1.14 | 1.15 | 0.12 | 0.01 | 0.00 | 0.00 | 0.95 | 1.00 | 0.99 | 0.99 |
| 2 | 0.6 | 20 | 8 | 1.21 | 1.13 | 1.12 | 1.14 | 0.12 | 0.00 | -0.00 | 0.00 | 0.91 | 0.99 | 0.94 | 0.97 |
| 3 | 0.4 | 15 | 5 | 1.19 | 1.12 | 1.11 | 1.14 | 0.11 | -0.00 | -0.01 | -0.01 | 0.87 | 0.96 | 0.87 | 0.90 |
| 5 | 0.3 | 10 | 3 | 1.17 | 1.10 | 1.09 | 1.13 | 0.11 | -0.01 | -0.03 | -0.02 | 0.78 | 0.90 | 0.67 | 0.72 |
| 10 | 0.2 | 5 | 2 | 1.14 | 1.05 | 1.05 | 1.12 | 0.09 | -0.04 | -0.12 | -0.04 | 0.57 | 0.68 | 0.20 | 0.50 |
| 15 | 0.1 | 2 | 1 | 1.11 | 0.97 | 1.00 | 1.12 | 0.08 | -0.16 | -0.52 | -0.12 | 0.39 | 0.16 | 0.01 | 0.08 |

-------------------------------------------------------------------------------------------------------

Quality Flag =1

--------------------

| | Range | | | Standard Deviations | | | | PDF Average | | | | Fraction of Data Points | | | |
|---|---|---|---|---|---|---|---|---|---|---|---|---|---|---|---|
| 0 | 1.0 | 40 | 20 | 3.91 | 3.64 | 3.53 | 3.60 | -1.07 | -0.95 | -0.94 | -0.96 | 1.00 | 1.00 | 1.00 | 1.00 |
| 1 | 0.8 | 30 | 10 | 3.20 | 3.54 | 3.45 | 3.47 | -0.69 | -0.93 | -0.94 | -0.95 | 0.80 | 0.95 | 0.94 | 0.94 |
| 2 | 0.6 | 20 | 8 | 2.88 | 3.31 | 3.26 | 3.40 | -0.53 | -0.80 | -0.89 | -0.93 | 0.68 | 0.86 | 0.80 | 0.90 |
| 3 | 0.4 | 15 | 5 | 2.68 | 2.94 | 3.12 | 3.22 | -0.42 | -0.56 | -0.85 | -0.87 | 0.58 | 0.72 | 0.66 | 0.76 |
| 5 | 0.3 | 10 | 3 | 2.49 | 2.77 | 2.96 | 3.04 | -0.32 | -0.49 | -0.84 | -0.79 | 0.45 | 0.59 | 0.43 | 0.54 |
| 10 | 0.2 | 5 | 2 | 2.27 | 2.75 | 3.27 | 2.92 | -0.28 | -0.55 | -1.32 | -0.75 | 0.27 | 0.35 | 0.11 | 0.35 |
| 15 | 0.1 | 2 | 1 | 2.13 | 3.47 | 4.88 | 2.93 | -0.26 | -1.25 | -2.74 | -0.86 | 0.16 | 0.07 | 0.00 | 0.06 |

1325 -------------------------------------------------------------------------------------------------------

[a] Columns 1-4 refer to Distkm, H(3D), H(Continuum), and CSNoiseRatio data screening
thresholds.



1330    Table 9. Multi-variable linear regression standard deviations and maxlatDiff values.[a]

| | Ocean, QF=0 | | | Ocean, QF=1 | | |
|---|---|---|---|---|---|---|
| Variable | Number | Stnd | maxlatDiff | Number | Stnd | maxlatDiff |
| Standard | 119144 | 0.86 | 0.46 | 53247 | 2.16 | 0.43 |
| | | | | 29434 | 1.41 | 0.55 |
| Distkm | 119144 | 0.85 | 0.41 | 53247 | 2.09 | 0.32 |
| | | | | 29434 | 1.39 | 0.51 |
| H(3D) | 119144 | 0.85 | 0.45 | 53247 | 2.13 | 0.41 |
| | | | | 29434 | 1.41 | 0.50 |
| CSNoiseRatio | 119144 | 0.84 | 0.39 | 53247 | 2.13 | 0.40 |
| | | | | 29434 | 1.39 | 0.47 |
| H(C) | 114137 | 0.85 | 0.46 | 53247 | 2.11 | 0.44 |
| | | | | 29434 | 1.40 | 0.53 |

| | Land, QF=0 | | | Land, QF=1 | | |
|---|---|---|---|---|---|---|
| Variable | Number | Stnd | maxlatDiff | Number | Stnd | maxlatDiff |
| Standard | 155602 | 1.24 | 0.09 | 113147 | 3.27 | 0.42 |
| | | | | 91620 | 2.75 | 0.34 |
| Distkm | 155602 | 1.24 | 0.08 | 113147 | 3.24 | 0.55 |
| | | | | 91620 | 2.73 | 0.43 |
| H(3D) | 154599 | 1.24 | 0.28 | 113044 | 3.23 | 0.39 |
| | | | | 91518 | 2.75 | 0.42 |
| CSNoiseRatio | 155602 | 1.24 | 0.09 | 113147 | 3.25 | 0.54 |
| | | | | 91620 | 2.74 | 0.49 |
| H(C) | 154582 | 1.23 | 0.10 | 112449 | 3.26 | 0.45 |
| | | | | 91064 | 2.74 | 0.30 |

[a]"Standard" refers to multiple linear regressions in which only the Version 10 standard variables (dPsco2, co2graddel for ocean; and dPfrac, CO2graddel, aodfine and log(DWS) for land) are utilized. The lower number in the QF=1 pairs refers to calculations with a restricted range of data (similar to that for the QF=0 data) for the standard variables.

Variable "Distkm" indicates taht the standard variables, plus the Distkm variable, are





used in the multiple-regression calculations. "Number" refers to the number of observations used in the calculations. H(C) refers to the H(Continuum) metric.

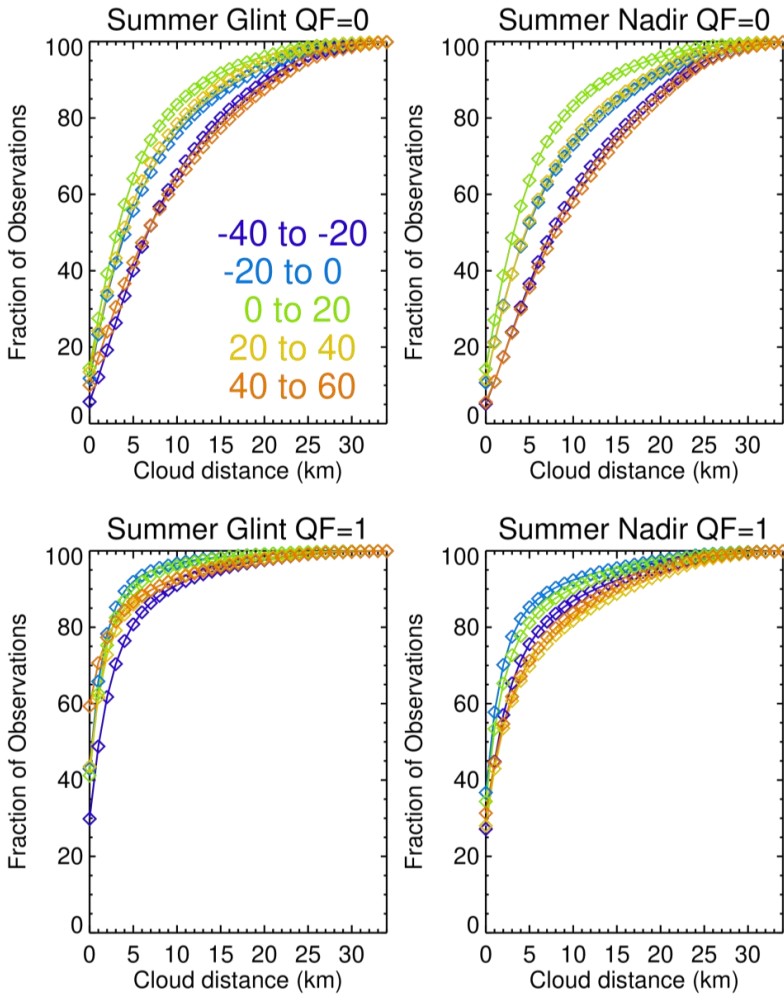

**Figure 1**. Fraction of observations that have a cloud within a circle of a specified radius (given by the x axis values) in summer for Ocean Glint and Land Nadir Lite file data points for QF=0 (best quality) and QF=1 (lesser quality) data. Each curve is for a labeled 20° latitudinal band. QF=1 fractions are generally larger than the QF=0 fractions.


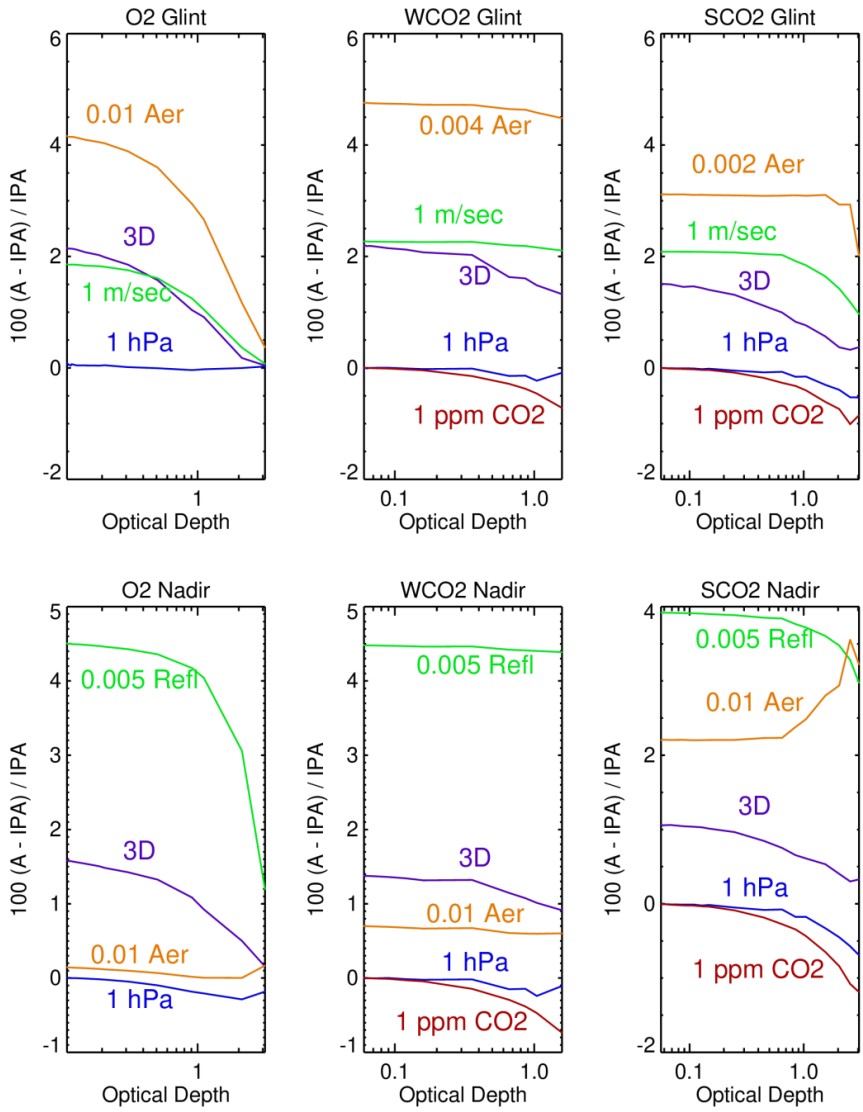

**Figure 2**. SHDOM 1D (IPA) and 3D radiative perturbations for ocean glint and land nadir viewing geometry using the same Fig. 9 cloud field. "A" in the y-axis title refers to 3D or 1D radiative perturbations. The sun is along the negative x-axis. The observation footprint is 4 km west from a cloud that is located at the x-y-z plane origin, corresponding to the June 12, 2016 cloud field observed by MODIS (see Fig. 9) over the ocean. Shadows are not located at this observation footprint since the sun and footprint are to the west of the cloud. The 3D radiance perturbations for glint viewing geometry are larger than the nadir viewing geometry perturbations.



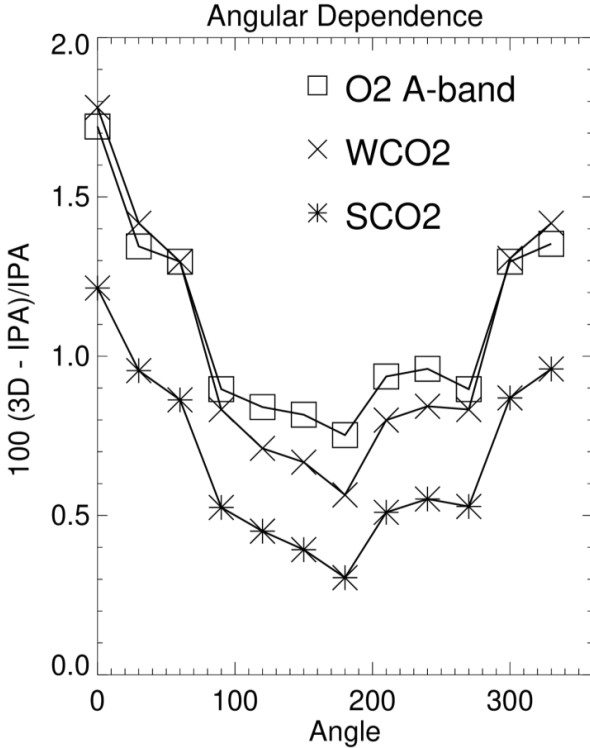

**Figure 3**. The angular dependence of the SHDOM 100 (3D-IPA)/IPA radiative
perturbations for glint view geometry for observation footprints along a circle 4 km
surrounding an isolated cloud. The observation footprints are to the west, north, east, and
south of the cloud at angles of 0°, 90°, 180° and 270°. The sun is along the –x axis and the
sensor is along the +x axis.

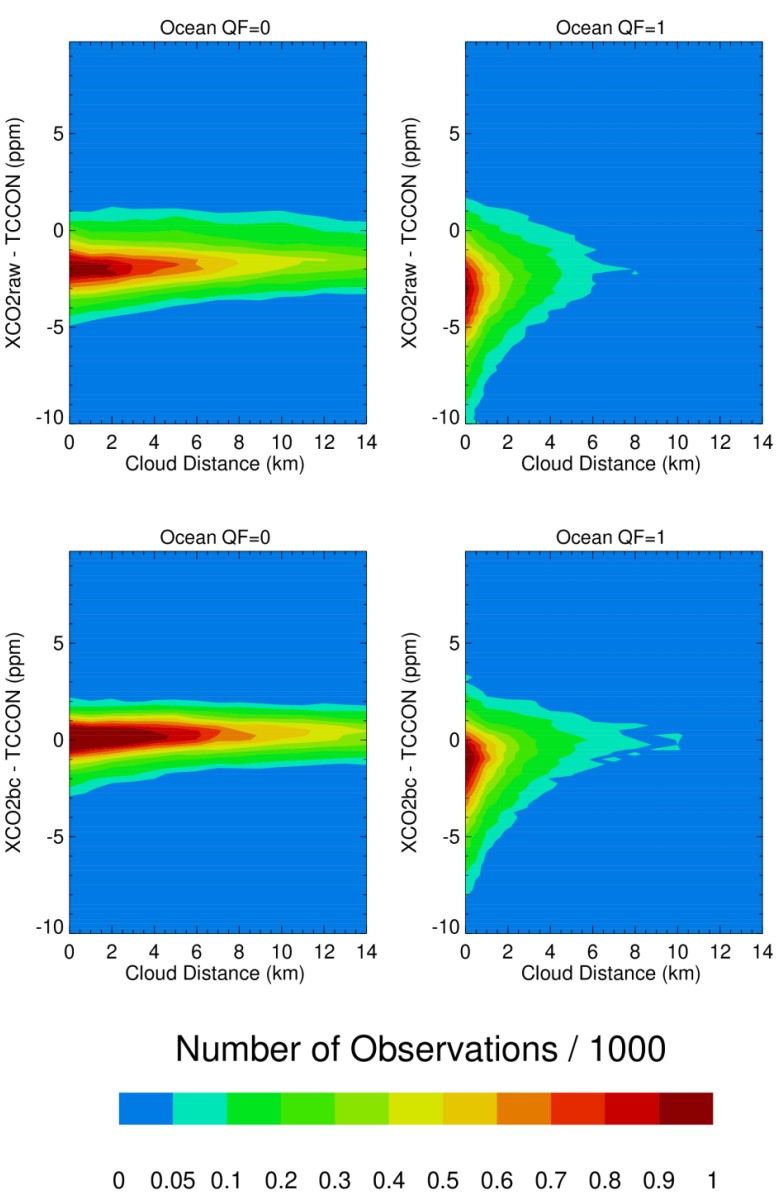

**Figure 4**. Contour maps of XCO2 – TCCON over the ocean as a function of the nearest cloud distance for QF=0 and QF=1 XCO2raw and XCO2bc Version 10 data. There is a very noticeable asymmetry (a tail of negative XCO2bc-TCCON) along vertical lines of nearest cloud distance in the QF=1 data, especially for small nearest cloud distance.



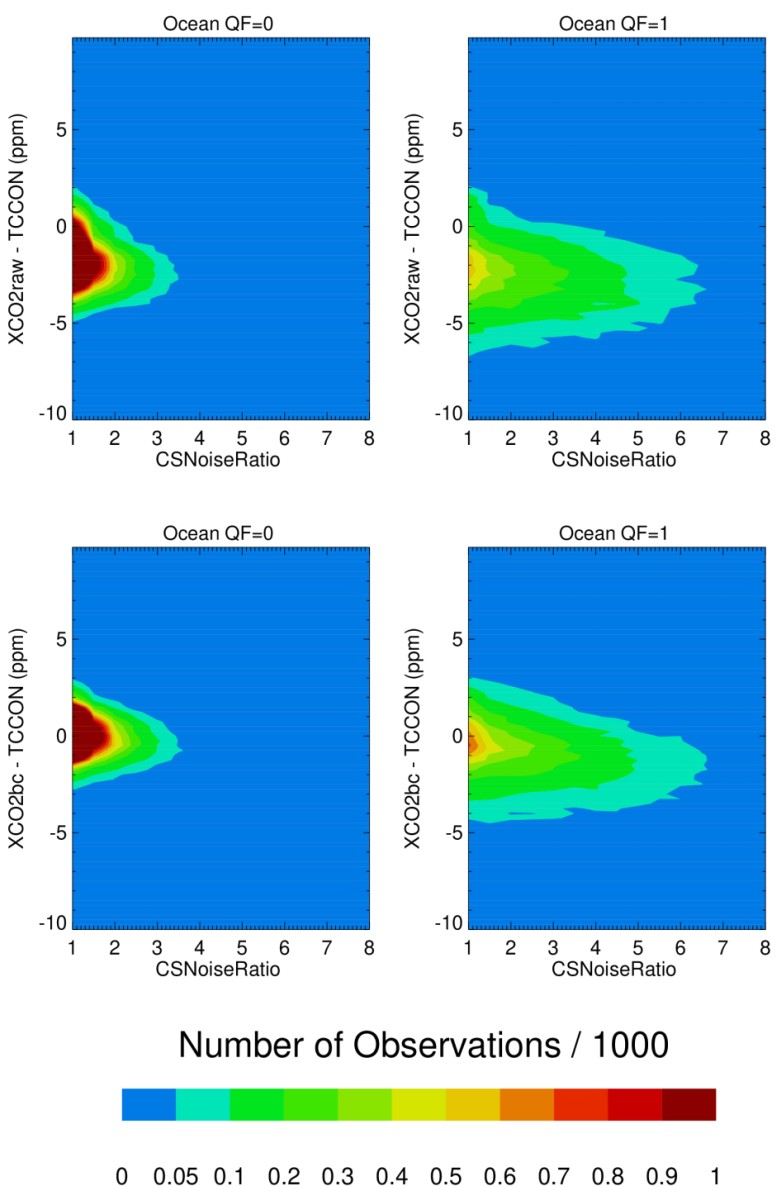

**Figure 5**. Contour maps of XCO2 – TCCON over the ocean as a function of the CSNoiseRatio metric for QF=0 and QF=1 XCO2raw and XCO2bc Version 10 data. The QF=1 XCO2bc data over the ocean has a noticeable asymmetry along CSNoiseRatio vertical lines.



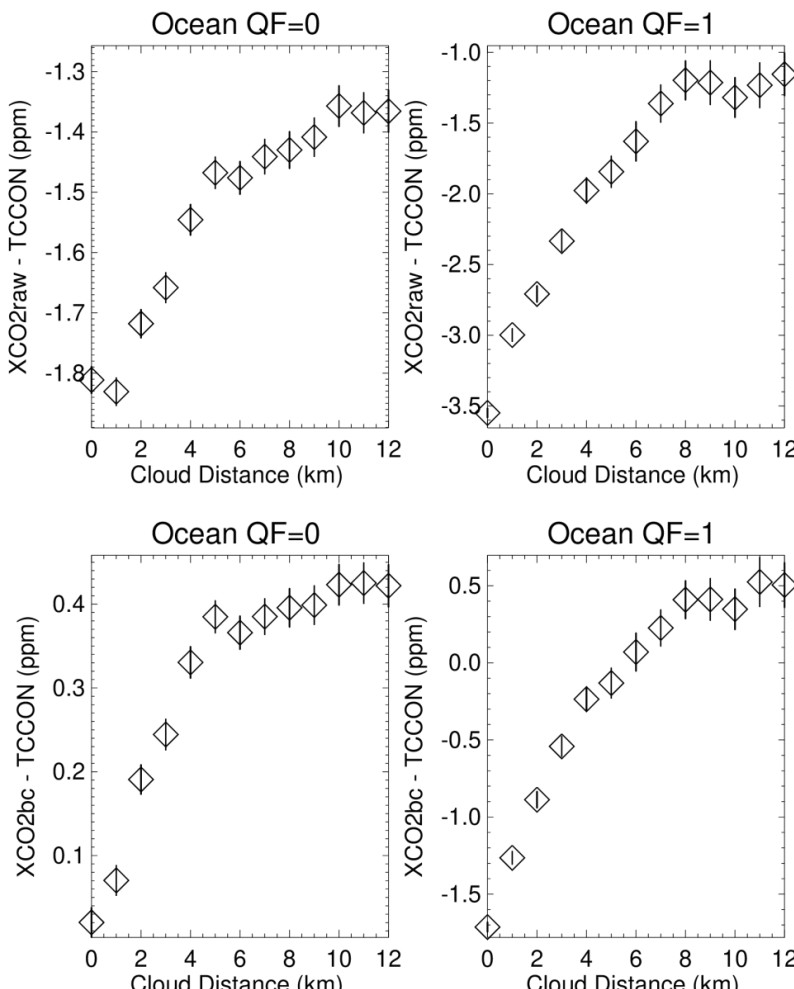

**Figure 6.** Averages of XCO2 – TCCON over the ocean as a function of the nearest cloud distance for QF=0 and QF=1 XCO2raw and XCO2bc Version 10 data. 95 % ($2\sigma$) confidence limits of the averages are represented by the vertical line associated with each average. The averages become more negative as the nearest cloud distance decreases. This indicates that the operational bias correction has a non-zero residual 3D cloud bias.



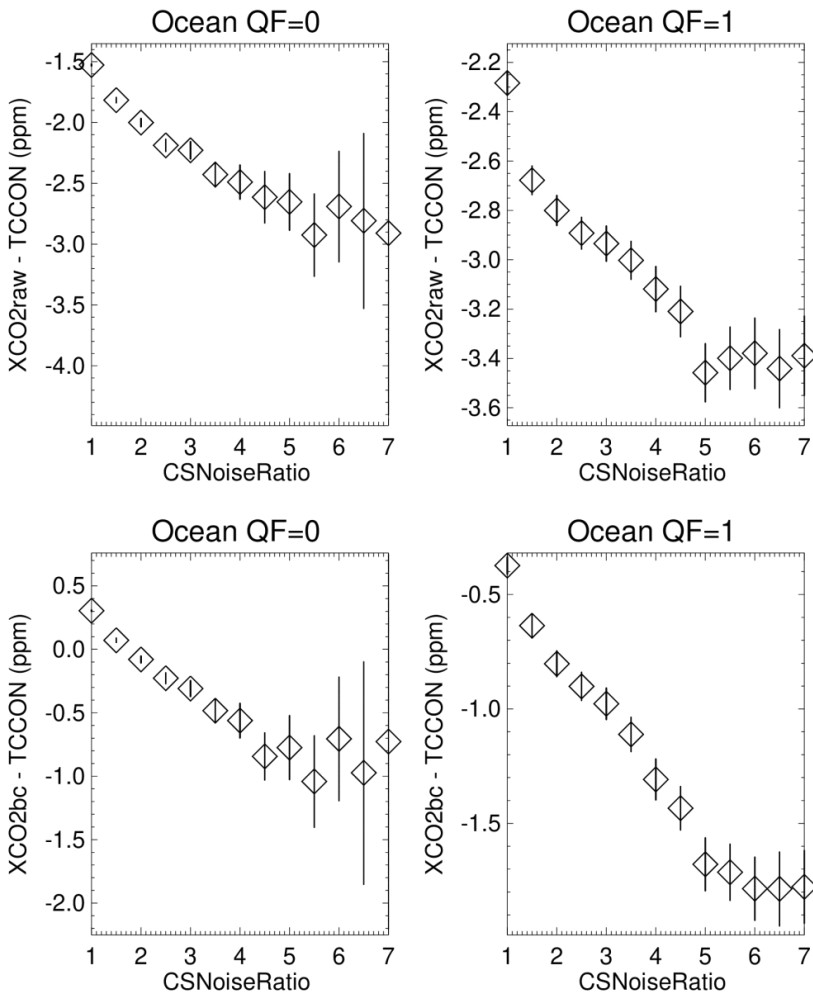


**Figure 7**. Averages of XCO2 – TCCON over the ocean as a function of the CSNoiseRatio metric for QF=0 and QF=1 XCO2raw and XCO2bc Version 10 data. 95 % (2σ) confidence limits of the averages are represented by the vertical line associated with each average. The averages become more negative for the QF=0 and QF=1 quality

flags as the CSNoiseRatio metric increases.



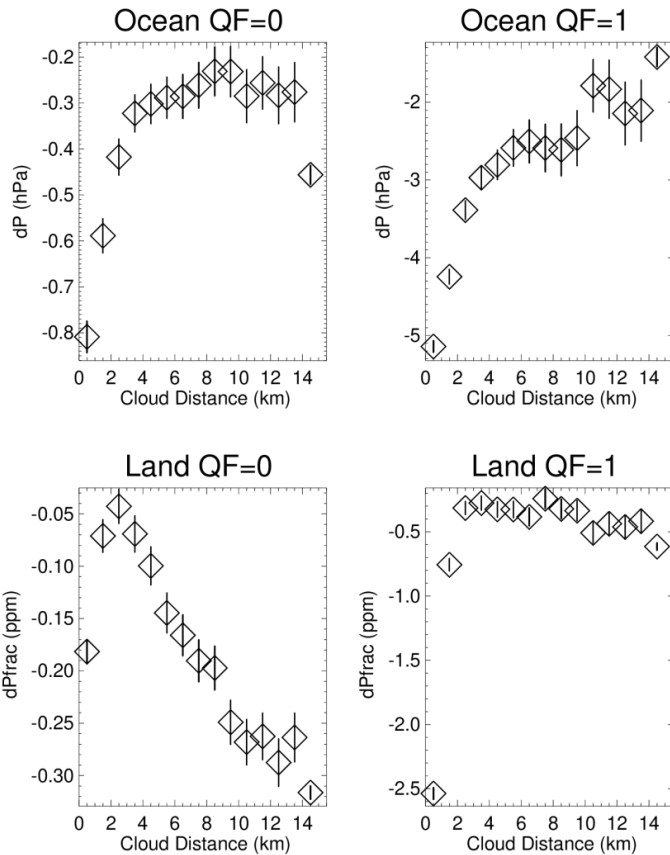

**Figure 8.** Averages of dPsco2 over the ocean and dPfrac over land as a function of the nearest cloud distance metric for QF=0 and QF=1 Version 10 data. 95 % (2σ) confidence limits of the averages are represented by the vertical line associated with each average.

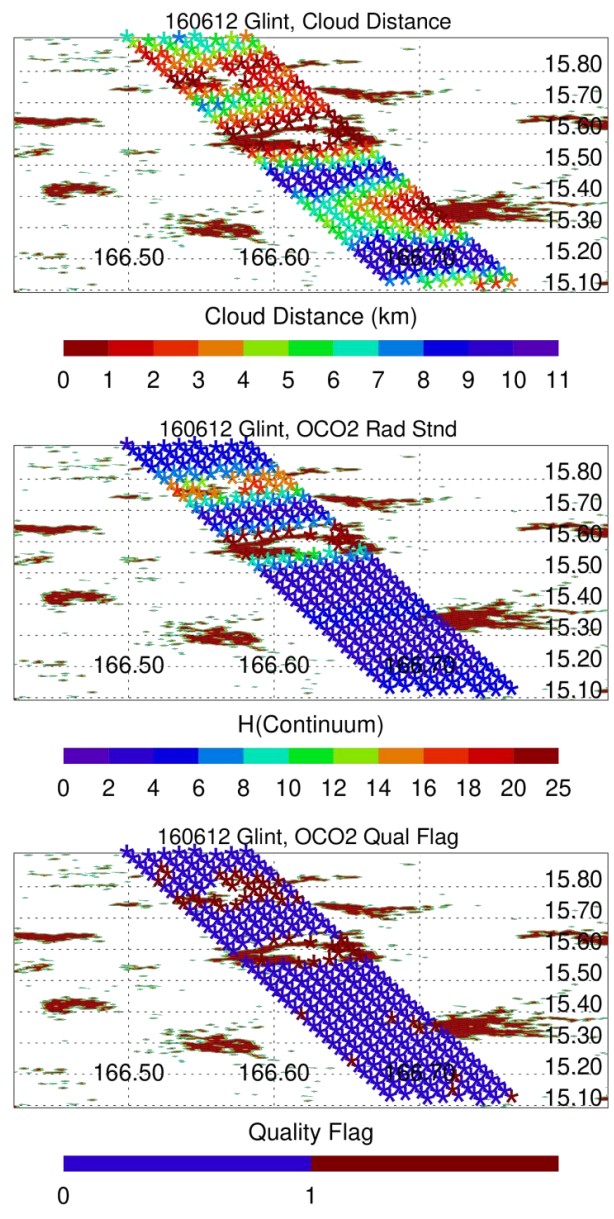


**Figure 9**. Geospatial variations in nearest cloud distance, O2-Aband continuum H(Continuum), and Quality Flag values for an ocean glint scene on June 12, 2016. Footprint observations are indicated by * symbols, and the MODIS cloud field is given by the irregular red shapes. Longitude and Latitude are given by the x and y axes.



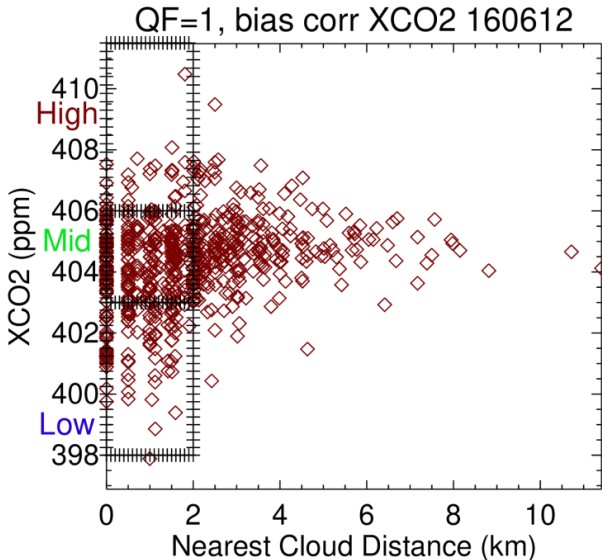

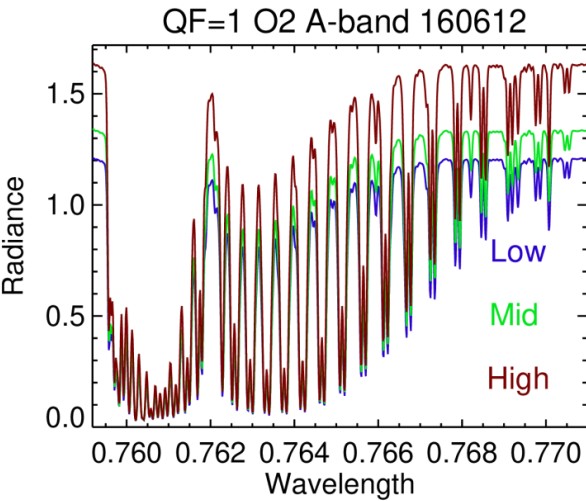

**Figure 10**. Bias corrected Version 10 XCO2bc versus nearest cloud distance for QF=1 data for a region that extends north and south of the June 12, 2016 scene illustrated in Fig. 9. The bottom panel presents $O_2$ A-band average spectra for the three boxes in the upper panel.


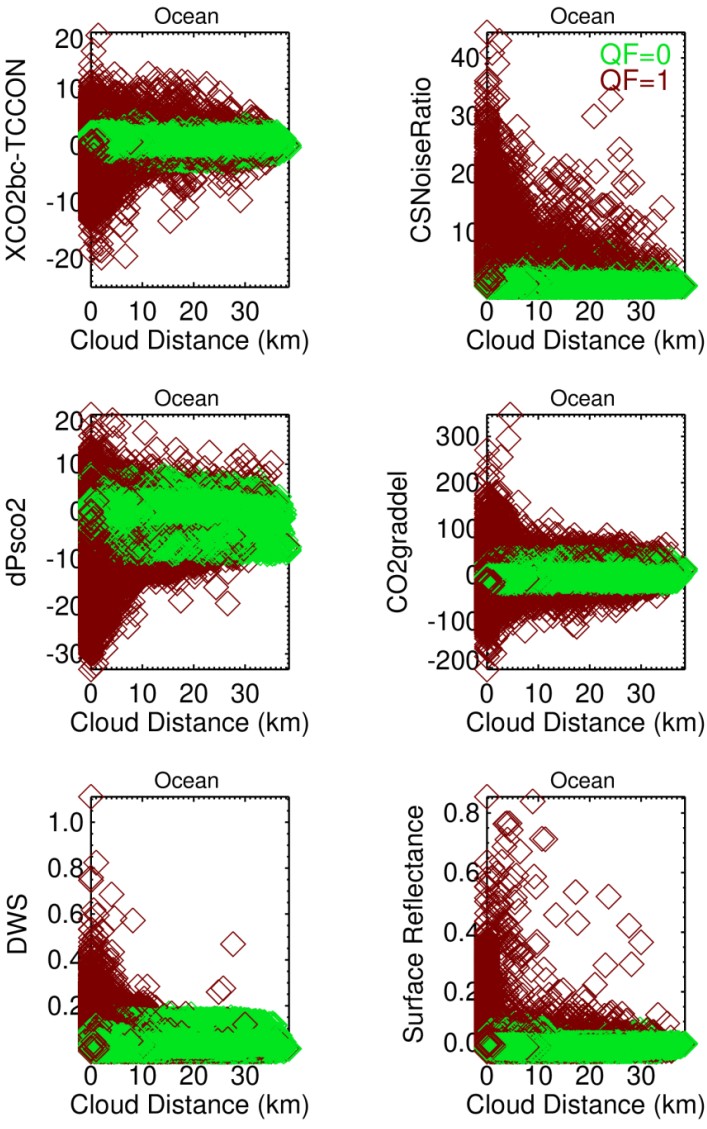

**Figure 11**. Dependence of Version 10 ocean bias correction variables (dPsco2, CO2graddel) and other variables (DWS, surface reflectance, and CSNoiseRatio) as a function of nearest cloud distance and Quality Flag data. The data points are for a limited range of latitude (52S ° - 41S°) and longitude (164° - 180°) in 2017.






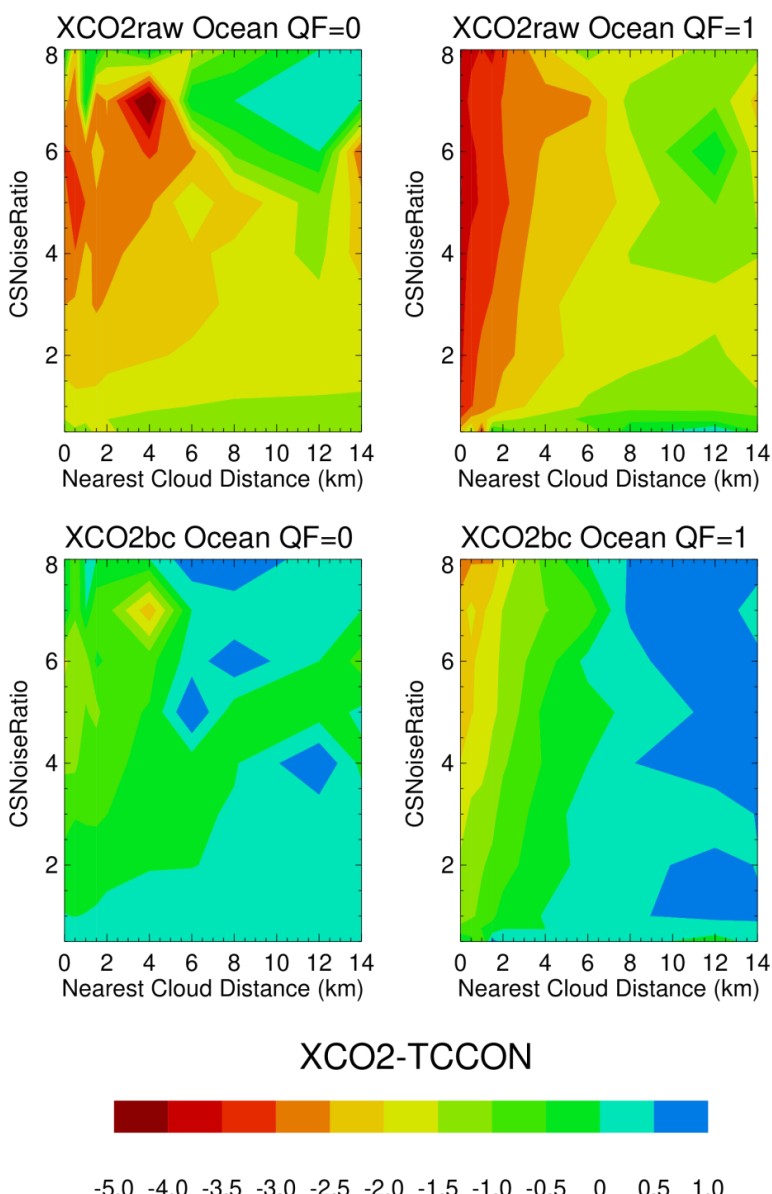

**Figure 12**. Contour graphs of XCO2raw-TCCON and XCO2bc-TCCON for ocean glint measurements. Largest differences are present at smallest nearest cloud distances and largest CSNoiseRatio values especially for the QF=1 data.


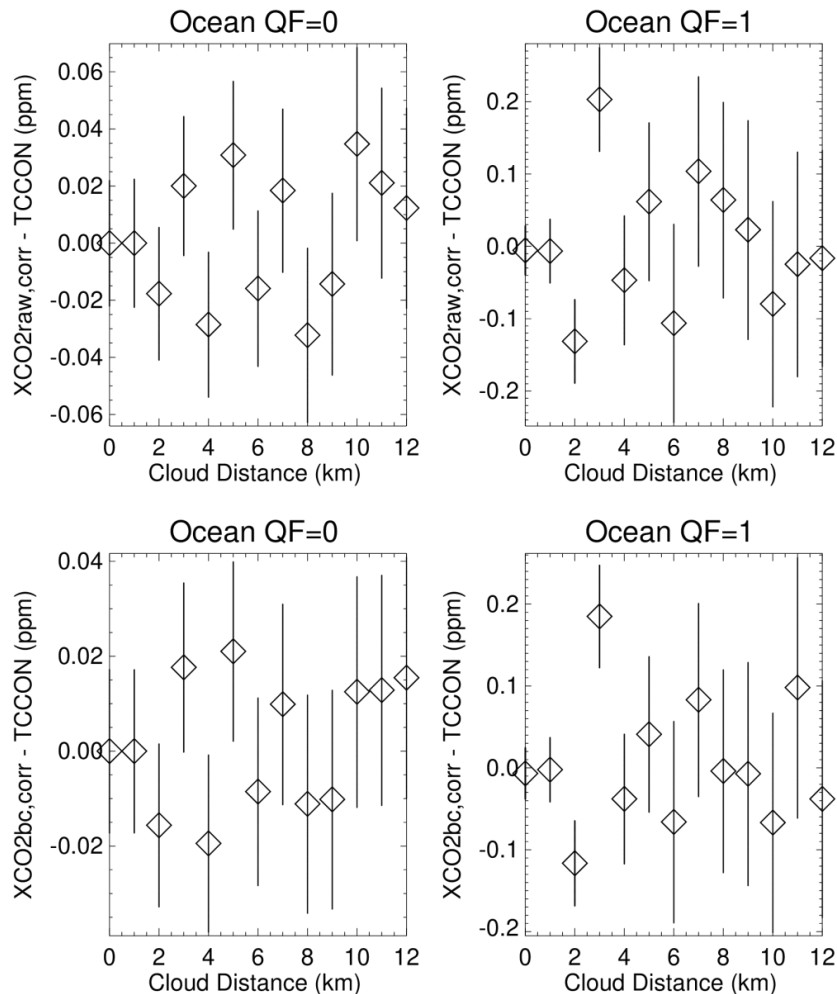

**Figure 13**. Application of Fig. 12, used as a Table look-up correction for 3D cloud biases, leads to revised XCO2raw,corr-TCCON and XCO2bc,corr-TCCON averages for ocean data, binned as a function of nearest cloud distance.




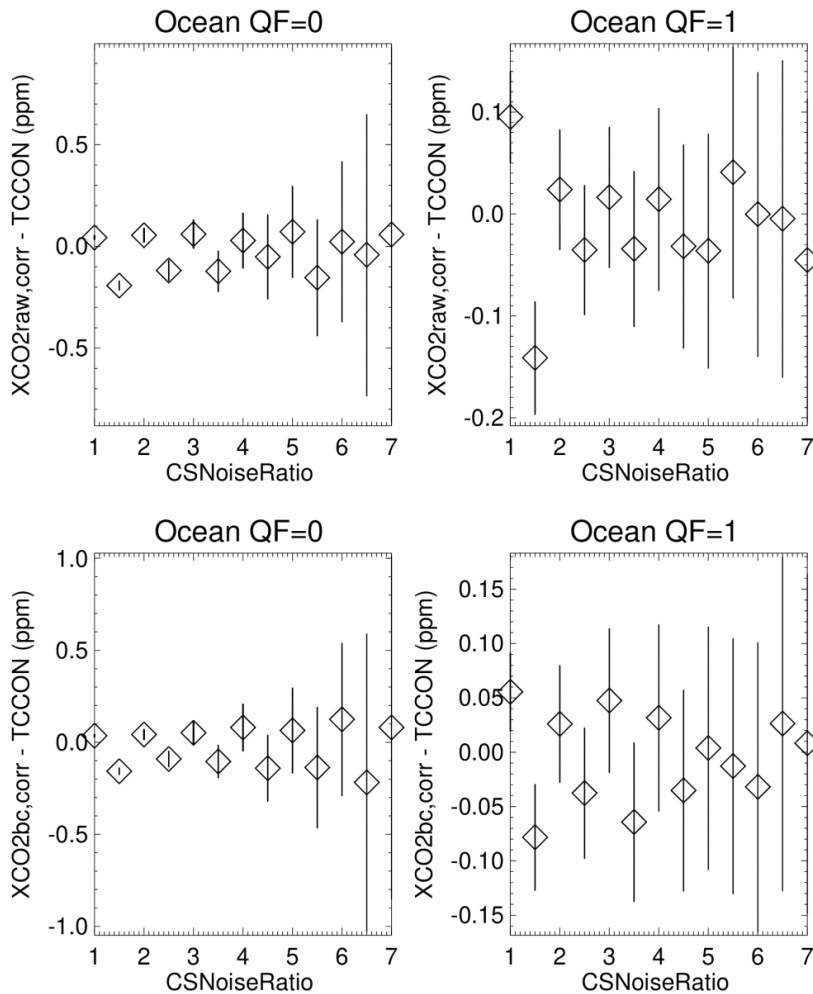

**Figure 14**. Application of Fig. 12, used as a Table look-up correction for 3D cloud biases, leads to revised XCO2raw,corr-TCCON and XCO2bc,corr-TCCON averages for ocean data, binned as a function of the CSNoiseRatio 3D metric.


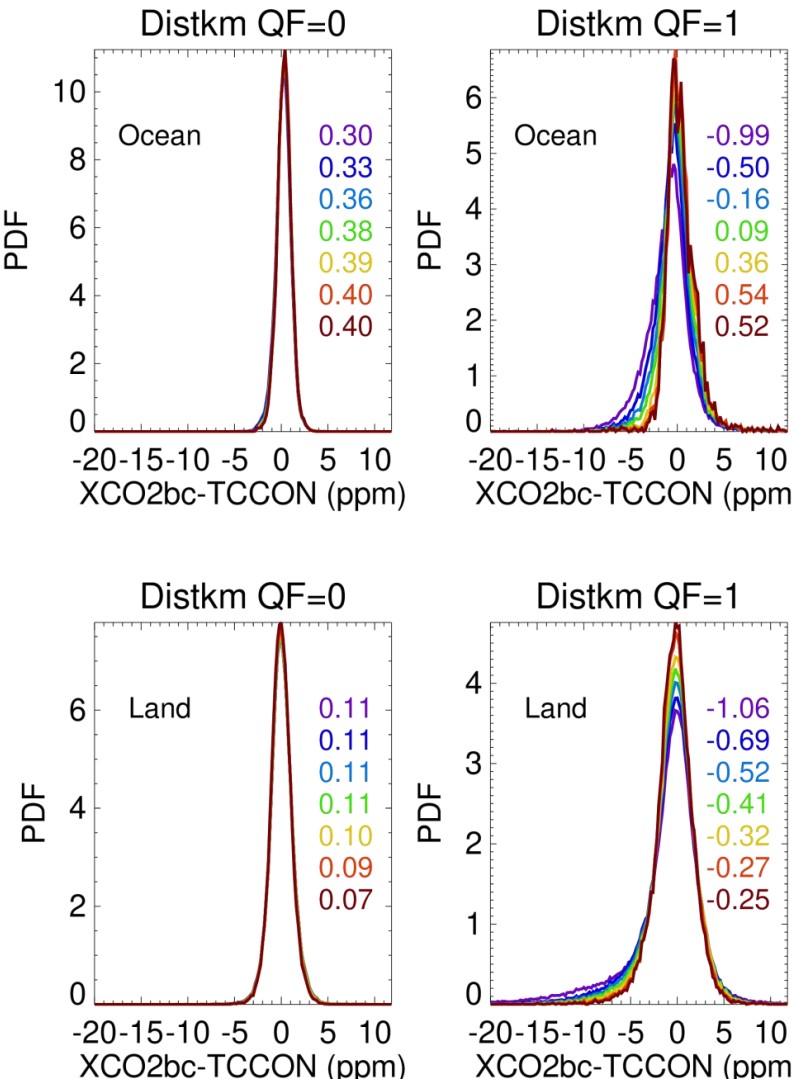

**Figure 15**. Changes in the PDFs of Version 10 XCO2bc-TCCON as a function of the nearest cloud distance screening process (see Tables 7 and 8). The numbers in the panels are the number weighted XCO2bc-TCCON averages (in ppm) of the PDFs, for nearest cloud screening threshold distances of 0, 1, 2, 3, 5, 10, and 15 km.
