# Peer review of "Analysis of 3D Cloud Effects in OCO-2 XCO2 Retrievals"

_Atmospheric Measurement Techniques, 2020_

## Referee Comment (RC1) · Anonymous Referee #1 · 18 Oct 2020

The manuscript describes 3D cloud effects in OCO-2 XCO2 retrievals. This is done using both measurements (TCCON, OCO-2 and MODIS) and 3D radiative transfer simulations. The presence of such effects are clearly demonstrated and their importance discussed. Various mitigation methods are presented and discussed.

The manuscript is well-organised and include detailed description of the results. It is recommended for publication after consideration of the minor comments below.

Comments

• **Table of acronyms**: The manuscript contains numerous acronyms. Some are self-explanatory, some common and some rather unusual in this context (like

DWS which made this reviewer think about deep water soloing). To help the reader, please include a table of all acronyms and their explanations.

- Page 2, line 47: Rayner and O'Brien (2001) is missing in the References.
- **Page 4, lines 140-183**: Please specify the OCO-2 pixels size. And please provide a rough number of how many MODIS pixels cover one OCO-2 pixels.
- Page 6, lines 249-251: This sentence is hard to read. Please rephrase.
- Page 6, line 263: Please explain what is meant by "eight OCO-2 observation footprints".
- **Page 8, lines 354-368**: Please include information about cloud phase (liquid or ice water cloud, I presume the former, but it should be written in the manuscript). How was the optical properties of the cloud calculated? What is the cloud effective radius and how was it estimated?

AMTD

---

## Referee Comment (RC2) · Anonymous Referee #2 · 4 Nov 2020

I believe this manuscript presents work that is worthy of publication. Most importantly, it estimates biases related to the presence of nearby clouds in satellite measurements of atmospheric carbon dioxide amounts. The methodology is reasonable and the presentation is generally good. Even so, I do have some significant concerns about the current version of the manuscript, and I recommend some major revisions. Please find my specific comments below.

Main issues:

1.

My main comment is about the attribution of retrieval biases to 3D radiative effects. I wonder if, in addition to 3D effects, other factors may also play significant roles in

the analyzed biases. It is clear that the biases are caused by factors and processes related to the presence of nearby clouds, but perhaps not exclusively by 3D radiative effects. I wonder mainly about two other cloud-related factors. First, the surroundings of detected clouds are likely to contain some undetected clouds as well (subpixel-size clouds, cloud fragments detrained from larger clouds, remnants of mostly dissipated clouds, etc.). Second, aerosol optical depth increases near clouds due to factors such as the hygroscopic swelling of aerosols caused by the increased near-cloud humidity. The manuscript should discuss at least briefly–or perhaps using some calculations–whether cloud contamination or aerosol swelling (or even just the increased near-cloud humidity) could also play a role in the analyzed biases. If so, the findings should probably be reframed in the title and throughout the manuscript.

2.

Section 11 describes various attempts to improve the accuracy of bias-removal methods, but the authors conclude that none of the attempts proved successful in the end. Because the manuscript is already quite long, I suggest reducing the length of the section and limiting it to only a few sentences saying that the authors tried these approaches, but they did not prove helpful. Perhaps these sentences could even be merged into some other section The details of the unsuccessful attempts do not seem critical and my sense is that even Table 9 could be deleted. In general, the number and size of tables is quite large, and if the authors found ways to delete some other tables–or at least to move them into an appendix or supplemental material–this could make the paper more inviting to readers.

3.

It seems that the procedure described in Lines 625-627 should be affected by the random sampling noise that appears to cause some small-scale variability (local minima or maxima at certain Distkm-CSNoiseRatio bins) in Figure 12. If the bias correction were to be applied to a different dataset (which has its own different sampling noise),

this small-scale noise would presumably introduce additional errors into the correction. In addition to various nonlinearities, this sampling noise might also be a factor in why (as mentioned in Lines 638-640) linear regression is not performing as well as the bin-based process (Lines 625-627) for this dataset. I believe the manuscript should discuss the topic of sampling noise/variability somewhere.

Other issues:

Line 56: The word "ratio" should be added after "signal to noise".

Line 155: It should be clarified where exactly the information contained in the CSU files comes from. Are these files created by combining selected data from operational MODIS products—and if so, which ones?

Lines 159-160: Does it ever occur that the MODIS cloud product retrieves a cloud optical depth greater than 1.0 and yet the MODIS cloud mask does not say the pixel is cloudy? If yes, it would be interesting to discuss when and why this happens. If not, the word "or" may have to be replaced by "and".

Line 263: It would help to clarify what happens if clouds occur inside the OCO-2 footprint.

Lines 283-284: For the benefit of readers not familiar with OCO-2, it would help to specify somewhere (in addition to the Crisp reference) what the OCO-2 pixel and footprint sizes are, what the difference is between the two, why 8 footprints are grouped together and how these footprints are arranged. Some of this is mentioned in Lines 298-299, but it would be helpful to see this (and the rest of the information) a bit earlier, right when first mentioned.

Lines 323-325: The wording should be refined to clarify whether land and ocean are combined or QF=0 and QF=1 are combined. In other words, whether the 40% is for QF=0 (land+ocean) and 73% is for QF=1 (land+ocean), or 40% is for land (QF=0 + QF=1) and 73% is for ocean (QF=0 + QF=1).

Table 2: It would help to clarify the used definition of seasons. For example, do summer statistics combine data from June-July-August over the Northern Hemisphere with data from December-January-February over the Southern Hemisphere?

Line 342: Shouldn't Figure 9 be moved to become Figure 2, just so readers don't need to jump from Figure 1 to Figure 9?

Line 349: It would help to specify what wavelength the monochromatic total optical depth is for.

Section 4: It would help to mention, if this is known, whether the key difference between the different 3D measures is that they consider standard deviation values over different spatial scales or at different wavelengths–or is it something else?

Figure 2 caption: The sentence "The sun is along the negative x axis" does not fit here; the x-axis shows optical depth, not any position or angle. The end portion of the caption also seems to refer to simulation setup and could be deleted, especially as the text mentions some of this info anyway (e.g., Line 382).

Lines 481-482: I recommend explaining why the 0.4 ppm bias at large distances from clouds can be attributed to 3D effects. This seems counter-intuitive, as this bias occurs in far-from-cloud cases where 3D effects should be weakest. Perhaps 3D effects that occur closer to clouds make the bias correction to be incorrect far from clouds? If the bias correction aims to remove overall biases (as mentioned in Lines 517-52), an overall correction that reduces biases near clouds could perhaps increase biases far from clouds at the same time?

Table 5 or other parts of Section 7: I wonder if the measures with the largest 3D biases are most suitable for capturing the key aspects of 3D effects, and measures with smaller biases are less so. In the extreme, an inept measure with no useful information about 3D effects would provide an estimate of zero for 3D effects. If this seems right, it may be worth mentioning in the paper.

Line 656: I guess it should be "5 and 10 km", not "5 and 50 km".

Line 765: The word "ocean" should be deleted.

Lines 777-778: It also seems potentially important and worth mentioning in the paper that clouds can move closer or farther as they drift with the wind during the 6 minutes between the OCO-2 and Aqua overpasses.

―――――――――――――――――

---

## Author Comment (AC1) · 9 Dec 2020

The manuscript describes 3D cloud effects in OCO-2 XCO2 retrievals. This is done using both measurements (TCCON, OCO-2 and MODIS) and 3D radiative transfer simulations. The presence of such effects are clearly demonstrated and their importance discussed. Various mitigation methods are presented and discussed. The manuscript is well-organized and include detailed description of the results. It is recommended for publication after consideration of the minor comments below.

Comments

Table of acronyms: The manuscript contains numerous acronyms. Some are self-explanatory, some common and some rather unusual in this context (like DWS which made this reviewer think about deep water soloing). To help the reader, please include a table of all acronyms and their explanations.

A table of acronyms is included in the revised paper (lines 909-965, revised paper line numbers).

Page 2, line 47: Rayner and O'Brien (2001) is missing in the References.

The Rayner and O'Brien reference is now included in the revised paper.

Page 4, lines 140-183: Please specify the OCO-2 pixels size. And please provide a rough number of how many MODIS pixels cover one OCO-2 pixels.

On lines 166-172 of the revised paper, these sentences were added:

For nadir view geometry, the OCO-2 footprint is approximately 1.3 km x 2.3 km at the Earth's surface (OCO-2 L2 ATBD, 2019). Eight adjacent footprints are arranged in a row (see Figure 2.2 of OCO-2 L2 ATBD, 2019), and these footprints in conjunction with the observation mode (ocean glint, land nadir, and target mode) determine the footprint scan patterns. Since the MODIS CSU radiances are archived at 500 m resolution, approximately 10 MODIS 500 m pixels fit within one OCO-2 footprint.

Page 6, lines 249-251: This sentence is hard to read. Please rephrase.

On lines 265-268 of the revised paper the revised sentences are:

Several 3D metrics are calculated from MODIS and OCO-2 data files. Nearest cloud distance (abbreviated as Distkm), the sun-cloud-footprint scattering angle, and the H(3D) metrics (discussed below) are calculated from MODIS data files. The CSNoiseRatio and the H(Continuum) metrics (discussed below) are calculated from *stand-alone* OCO-2 data.

Page 6, line 263: Please explain what is meant by "eight OCO-2 observation footprints".

The line has been revised to (lines 277-281):

The Distkm metric frequently refers to clouds that are *outside* of the geospatial scan pattern defined by the OCO-2 observation footprints. A representative scan pattern is illustrated in Figure 9, for glint (ocean) scene. There are clouds within and outside of the geospatial scan pattern marked by the asterisks.

Page 8, lines 354-368: Please include information about cloud phase (liquid or ice water cloud, I presume the former, but it should be written in the manuscript). How was the optical properties of the cloud calculated? What is the cloud effective radius and how was it estimated?

On lines 394-403 the following paragraph was added to the revised paper:

A separate computer program calculates the three dimensional distribution of water droplets and aerosol particles in the x-y-z grid, writing to an offline data file. This file specifies the liquid water contents and effective radii of the water droplets, and the aerosol mass densities and effective radii. We specified water droplets to have an effective radius of 10 μm, and aerosol particles an effective radius of 0.1 μm. SHDOM uses a Mie calculation to write to a particle scattering table for a range of water droplet effective radii (for a gamma size distribution), and a similar table for the aerosol particles (for a lognormal size distribution). These two tables, and the offline input file, are used by SHDOM to specify the particle absorption, scattering, and phase function particle characteristics in the x-y-z grid.

---

## Author Comment (AC2) · 9 Dec 2020

I believe this manuscript presents work that is worthy of publication. Most importantly, it estimates biases related to the presence of nearby clouds in satellite measurements of atmospheric carbon dioxide amounts. The methodology is reasonable and the presentation is generally good. Even so, I do have some significant concerns about the current version of the manuscript, and I recommend some major revisions. Please find my specific comments below.

Main issues:

1.
My main comment is about the attribution of retrieval biases to 3D radiative effects. I wonder if, in addition to 3D effects, other factors may also play significant roles in the analyzed biases. It is clear that the biases are caused by factors and processes related to the presence of nearby clouds, but perhaps not exclusively by 3D radiative effects. I wonder mainly about two other cloud-related factors. First, the surroundings of detected clouds are likely to contain some undetected clouds as well (subpixel-size clouds, cloud fragments detrained from larger clouds, remnants of mostly dissipated clouds, etc.). Second, aerosol optical depth increases near clouds due to factors such as the hygroscopic swelling of aerosols caused by the increased near-cloud humidity. The manuscript should discuss at least briefly–or perhaps using some calculations– whether cloud contamination or aerosol swelling (or even just the increased near-cloud humidity) could also play a role in the analyzed biases. If so, the findings should probably be reframed in the title and throughout the manuscript.

> Yes, cloud fragments and hygroscopic swelling of aerosols caused by increased near-cloud humidity are real physical effects. The situations for which OCO-2 is most likely susceptible to 3D cloud effects are for low altitude "popcorn cloud" fields. A clear sky footprint, accompanied by an isolated cloud several km from the footprint, is a scene that passes the OCO-2 pre-screening tests. Scenes in which there are substantial clouds very close to the footprint are scenes rejected by the cloud pre-screener. If there were a cloud fragment close to a clear sky footprint embedded in a popcorn cloud field, then it would introduce optical depth to the scene and influence the 3D radiative transfer.

> We used Google Scholar to search for papers on cloud fragments and increased near-cloud humidity. We found articles on the latter, but not the former topic.

There are observation and modeling papers on increased near-cloud humidity. Twohy et al (Twohy, C. H., J. A. Coakley Jr., and W. R. Tahnk (2009), Effect of changes in relative humidity on aerosol scattering near clouds, J. Geophys. Res., 114, D05205, doi:10.1029/2008JD010991 ) measured relative humidity and aerosol scattering in the vicinity of small marine cumulus during the 1999 Indian Ocean Experiment (INDOEX). Relative humidity increased as distance to the boundaries of small marine trade cumulus decreased.

From their Figure 4, the near-cloud humidity increase occurs within 1 km of the clouds they observed.

[Figure]

**Figure 4.** Average departures from the means for flight legs entering clouds for the cloud-free portions composited for all INDOEX flight legs encountering low-level clouds. The departures are for (a) relative humidity, (b) particle concentrations from the PCASP-100, (c) CN counter, and (d) FSSP-300. Average RH for the flight legs was 88–90%. Average particle concentrations were 900 cm$^{-3}$ for the PCASP, 2500 cm$^{-3}$ for the CN counter, and 30–40 cm$^{-3}$ for the FSSP-300. The distances are from the cloud edge. Separate lines show values for different lengths of cloud-free air prior to cloud edge ranging from ~110 m (All) to 4 km.

The literature also contains papers in which these effects are modeled, with

Lu, M.-L., R. A. McClatchey, and J. H. Seinfeld (2002), Cloud halos: Numerical simulation of dynamical structure and radiative impact, J. Appl. Meteorol., 41, 832 – 848.

and

Lu, M.-L., J. Wang, A. Freedman, H. H. Jonsson, R. C. Flagan, R. A. McClatchey, and J. H. Seinfeld (2003), Analysis of humidity halos around trade wind cumulus clouds, J. Atmos. Sci., 60, 1041 – 1059.

representing two examples.

Figure 4 of Lu et al (2002) has the following model field.

[Figure]

FIG. 4. Simulated development of shallow cumulus off the northern CA coast near Oakland in the absence of wind, for several selected time stages. The contour represents a cloud LWC of 0.01 g m$^{-3}$, chosen to define the cloud boundary. The shaded region outside the cloud boundary, defined to be the cloud halo, is where the absolute humidity exceeds the e-folding water vapor density (see Fig. 1).

The cloud halo is on the order of ½ km from the main cloud features.

From Figure 6 of our paper, the XCO2bc-TCCON effect extends over a spatial scale of 10 km. This spatial scale is larger than the 1 km cloud halo spatial scale.

Lines 533-546 of the revised paper now discusses the issue of cloud fragments and cloud haloes, and the interpretation of the OCO-2 data:

The data presented in Fig. 6 and elsewhere in this paper could also be influenced by the presence of undetected cloud fragments, dissipating clouds, and the fact that relative humidity is enhanced directly outside a cloud. The increase in relative humidity leads to swelling of aerosols, which would enhance near-cloud aerosol scattering. Twohy et al. (2009) measured relative humidity and aerosol scattering in the vicinity of small marine cumulus during the 1999 Indian Ocean Experiment (INDOEX). Enhancements were observed within 1 km of the cloud. Observations and model simulations of "cloud haloes" by Lu et al. (2002) and Lu et al. (2003) also indicate that the cloud halo exists ~ ½ km from a cloud. From Fig. 6, however, the XCO2bc-TCCON averages asymptote to a constant value over a length scale of 10 km, a scale substantially larger than the 1 km scale associated with cloud haloes. This disfavors an interpretation that the variation in Fig. 6 is primarily due to cloud halo effects. Várnai and Marshak (2009) also concluded that aerosol swelling does not account for observed illuminated / shadowy asymmetries in MODIS shortwave reflectance versus nearest cloud distance data.

2.

 Section 11 describes various attempts to improve the accuracy of bias-removal methods, but the authors conclude that none of the attempts proved successful in the end. Because the manuscript is already quite long, I suggest reducing the length of the section and limiting it to only a few sentences saying that the authors tried these approaches, but they did not prove helpful. Perhaps these sentences could even be merged into some other section. The details of the unsuccessful attempts do not seem critical and my sense is that even Table 9 could be deleted. In general, the number and size of tables is quite large, and if the authors found ways to delete some other tables–or at least to move them into an appendix or supplemental material–this could make the paper more inviting to readers.

> We have deleted a full paragraph in Section 11, and have also deleted a paragraph in Section 12, in the revised text.

> We do think discussion of this mitigation technique in Section 11 is warranted since the "adding terms to the bias correction equations" is an obvious mitigation technique to try. If the paper did not include this technique, most readers would ask "why didn't you add the nearest cloud distance term to the bias correction equation? Not trying this technique is perplexing."

3.

It seems that the procedure described in Lines 625-627 should be affected by the random sampling noise that appears to cause some small-scale variability (local minima or maxima at certain Distkm-CSNoiseRatio bins) in Figure 12. If the bias correction were to be applied to a different dataset (which has its own different sampling noise), this small-scale noise would presumably introduce additional errors into the correction. In addition to various nonlinearities, this sampling noise might also be a factor in why (as mentioned in Lines 638-640) linear regression is not performing as well as the bin-based process (Lines 625-627) for this dataset. I believe the manuscript should discuss the topic of sampling noise/variability somewhere.

We have tried to improve upon the small-scale variability in Figure 12, by several refinements, but were unsuccessful in improving upon Figures 13 and 14. We then thought it best to present Figure 12, and apply it, without refinements. The main take-home message of the paper is a Table Look-up technique, utilizing two 3D metrics, yielded better results than other attempted techniques.

The operational retrieval and post-retrieval bias correction processing yields XCO2bc PDFs with substantial standard deviations (on the order of 0.8 ppm) even for clear sky conditions. The standard deviations increase when 3D cloud radiative effects are added to the spectra. The 3D cloud effects are embedded in a sea of complicated "retrieval code responses". So in addition to measurement noise in the OCO-2 spectra, there is noise associated with the retrieval code response to a radiance perturbation that is not physically described by the retrieval code physics.

Lines 868-872 were added to the text to discuss the noise/variability issue in general terms:

The Table Look-up technique is based upon data (see Figure 12) that has bin to bin variations. Some of the data bins in fact have zero input data points. The bin to bin variability introduces some noise to the correction process. Some of the bin to bin variation is likely due to the fact that the retrieval code response to radiative perturbations, for physics not included in the retrieval physics, is complicated and noisy.

Other issues:

Line 56: The word "ratio" should be added after "signal to noise".

Line 56 now reads "the signal to noise ratio"

Line 155: It should be clarified where exactly the information contained in the CSU files comes from. Are these files created by combining selected data from operational MODIS products and if so, which ones?

Lines 155-157 now includes the sentence:

Input to these auxiliary files include MODIS 1km MYD03 geolocation, 500 m MYD02HKM radiance files, and 1 km MYD06 cloud files, which includes the 1 km MODIS cloud mask.

Lines 159-160: Does it ever occur that the MODIS cloud product retrieves a cloud optical depth greater than 1.0 and yet the MODIS cloud mask does not say the pixel is cloudy? If yes, it would be interesting to discuss when and why this happens. If not, the word "or" may have to be replaced by "and".

As indicated in original paper lines 159-160, we identify a cloud if the MODIS cloud mask says a cloud is present or if the MODIS cloud optical depth is greater than 1.0. This optical depth detection threshold was determined empirically by co-author Dr. Sebastian Schmidt from his previous experience with MODIS data.

To answer your question, we downloaded MODIS MYD06 cloud and MYD35 cloud mask files from the NASA GES DISC website for June 12, 2016, since the original CSU files, based upon Dr. Cronk's v9 MODIS files, were completely scrubbed from the JPL computers to make room for new v10 MODIS files (which have not been created for the various types of MODIS files). For the June 12, 2016 date, 14% of the data points had the cloud optical depth greater than 1.0 while the MODIS cloud mask said that a cloud was not present. We don't know the MODIS team processing details that leads to this difference.

Since our calculations used the "or" case, the paper should say "or" because that is what we did.

Line 263: It would help to clarify what happens if clouds occur inside the OCO-2 footprint.

Line 281-283 was added to the text:

If a cloud is inside a footprint, then the cloud would add photons to the sensed radiance, and any cloud shadows would provide lesser sensed radiance

Lines 283-284: For the benefit of readers not familiar with OCO-2, it would help to specify somewhere (in addition to the Crisp reference) what the OCO-2 pixel and footprint sizes are, what the difference is between the two, why 8 footprints are grouped together and how these footprints are arranged. Some of this is mentioned in Lines 298-299, but it would be helpful to see this (and the rest of the information) a bit earlier, right when first mentioned.

Lines 166 now has been expanded (as suggested by the first reviewer):

For nadir view geometry, the OCO-2 footprint is approximately 1.3 km x 2.3 km at the Earth's surface (OCO-2 L2 ATBD, 2019). Eight adjacent footprints are arranged in a row (see Figure 2.2 of OCO-2 L2 ATBD, 2019), and these footprints in conjunction with the observation mode (ocean glint, land nadir, and target mode) determine the footprint scan patterns. Since the MODIS CSU radiances are archived at 500 m resolution, approximately 10 MODIS 500 m pixels fit within one OCO-2 footprint.

Lines 323-325: The wording should be refined to clarify whether land and ocean are combined or QF=0 and QF=1 are combined. In other words, whether the 40% is for QF=0 (land+ocean) and 73% is for QF=1 (land+ocean), or 40% is for land (QF=0 + QF=1) and 73% is for ocean (QF=0 + QF=1).

Lines 348-349 now read:

In approximate terms, 40 % (QF=0, glint or nadir) and 73 % (QF=1, glint or nadir) of the observations are within 4 km of clouds.

Table 2: It would help to clarify the used definition of seasons. For example, do summer statistics combine data from June-July-August over the Northern Hemisphere with data from December-January-February over the Southern Hemisphere?

The seasons are now defined in Table 2.

Line 342: Shouldn't Figure 9 be moved to become Figure 2, just so readers don't need to jump from Figure 1 to Figure 9?

Figure 9 could be moved to near line 342, but then the reader would need to junp back repeatedly to the Figure in Section 8, which focuses upon Figure 9. We think it will be less jarring to the reader to keep Figure 9 in Section 8.

An early reference to the OCO-2 footprint array is now given on lines 168-169:

Eight adjacent footprints are arranged in a row (see Figure 2.2 of OCO-2 L2 ATBD, 2019),…

Line 349: It would help to specify what wavelength the monochromatic total optical depth is for.

Lines 374 now reads:

with monochromatic total optical depths at representative wavelengths on the x axis and radiative perturbations on the y axis.

Section 4: It would help to mention, if this is known, whether the key difference between the different 3D measures is that they consider standard deviation values over different spatial scales or at different wavelengths–or is it something else?

On lines 336-341:

Of the four metrics, the nearest cloud metric is directly tied physically to the cloud field of a given scene, and is assessed over a wide spatial scale. The radiance inhomogeneity (radiance standard deviation) based metrics are indirectly tied to the cloud field, with the CSNoiseRatio and H(Continuum) metrics assessed over a small spatial range. We note, however, that a cloud field usually has more than one cloud, so the nearest cloud metric incompletely describes the cloud field.

The Figure 2. caption now reads:

**Figure 2**. SHDOM 1D (IPA) and 3D radiative perturbations for ocean glint and land nadir viewing geometry using the same Fig. 9 cloud field. "A" in the y-axis title refers to 3D or 1D radiative perturbations. The 3D radiance perturbations for glint viewing geometry are larger than the nadir viewing geometry perturbations.

Lines 481-482: I recommend explaining why the 0.4 ppm bias at large distances from clouds can be attributed to 3D effects. This seems counter-intuitive, as this bias occurs in far-from-cloud cases where 3D effects should be weakest. Perhaps 3D effects that occur closer to clouds make the bias correction to be incorrect far from clouds? If the bias correction aims to remove overall biases (as mentioned in Lines 517-52), an overall correction that reduces biases near clouds could perhaps increase biases far from clouds at the same time? Table 5 or other parts of Section 7: I wonder if the measures with the largest 3D biases are most suitable for capturing the key aspects of 3D effects, and measures with smaller biases are less so. In the extreme, an inept measure with no useful information about 3D effects would provide an estimate of zero for 3D effects. If this seems right, it may be worth mentioning in the paper.

The reviewer is correct to expect that the XCO2bc – TCCON averages should be close to zero at the largest cloud distances, since the 3D effect should physically asymptote towards zero as cloud distance becomes very large. The calculations in our paper, however, examine only XCO2bc – TCCON differences. The operational bias correction process looks at XCO2raw – TCCON and XCO2raw – model differences (from an ensemble of six models), and XCO2raw – small area analysis XCO2 (see added paragraph below). The final operational XCO2bc values are derived from a combination of the three comparisons. For this reason, our XCO2bc – TCCON averages are not equal to zero at large cloud distances. We choose to focus on XCO2bc – TCCON in our calculations, since TCCON XCO2 provides the most direct "truth proxy".

A paragraph has been rewritten in Section 7 (new lines 505-521):

Further insight into the Fig. 4 and 5 distributions is presented in Fig. 6 and 7, in which averages and 95 % (2σ) confidence limits of the averages are displayed. The XCO2raw-TCCON and XCO2bc-TCCON averages become more negative for both QF=0 and QF=1 cases as cloud distance approaches zero in Fig. 6. The averages become closer to each other as nearest cloud distance increases to large values. Ideally, the XCO2bc-TCCON differences should approach zero as the nearest cloud distance becomes very large, since the 3D effect should physically decrease towards zero as cloud distance becomes very large. The differences are close to 0.4 ppm in Fig. 6 instead of zero since the operational bias correction processing also considers comparisons to modeled XCO2 and small-area analysis in the determination of XCO2bc (O'Dell et al. 2018). Since the 95 % confidence limits in Fig. 6 do not overlap for small cloud distances, the differences in the averages, and the increasingly negative trend in the averages as cloud distance approaches zero, are statistically significant. This indicates that the operational bias correction does not completely remove 3D cloud effects from the XCO2raw retrievals for the full range of cloud distance. Fig. 6 indicates that there is a difference in the XCO2bc – TCCON averages

near -0.4 ppm (the difference of 0 ppm at cloud distances near 0 km and 0.4 ppm at cloud distances greater than 10 km). This difference is referred to as the ocean 3D *cloud bias*.

To expand upon the discussion of the use of model XCO2 data, we revised the first sentence of section 3 ( lines 195-199):

As discussed by O'Dell et al. (2018) and in the Version 9 OCO-2 Data Product User's Guide (2018, see Table 3.4), the bias correction procedure compares Level 2 retrieved XCO2raw to TCCON XCO2, model mean XCO2, and small area analysis XCO2 and produces *bias corrected* XCO2bc values, based upon the following equations for ocean glint and land nadir Version 9 observations.

and added this paragraph (lines 224-231):

As discussed by O'Dell et al. (2018), the small area analysis XCO2 is based upon the assumption that XCO2 should be uniform in a 100 km by 100 km region, since the XCO2 decorrelation length is between 500 and 1000 km. The model median data is taken from an ensemble of six models. The Feats coefficients are determined from a comparison of Feats coefficients derived separately from comparisons of XCO2raw with TCCON, model mean XCO2, and small area analysis XCO2. The TCCONadj divisor is based solely on TCCON data. In this paper we focus upon analysis of XCO2 –TCCON data since the TCCON data is the most direct truth proxy of the three proxies.

Line 656: I guess it should be "5 and 10 km", not "5 and 50 km".

Though Figure 1 has an x scale between 0 and 30 km, the processing of the MODIS CSU files yields Distkm values in the 0 to 50 km range.

Line 765: The word "ocean" should be deleted.

"ocean" has been deleted

Lines 777-778: It also seems potentially important and worth mentioning in the paper that clouds can move closer or farther as they drift with the wind during the 6 minutes between the OCO-2 and Aqua overpasses.

On line 817, the revised text now includes the sentence:

For a representative wind speed of 5 m/s, a cloud moves 1.8 km in six minutes, which is similar to the size of an OCO-2 footprint.